# EFFIBENCH-X: A Multi-Language Benchmark for Measuring Efficiency of LLM-Generated Code

**Yuhao Qing[1][*]**    **Boyu Zhu[2][*]**    **Mingzhe Du[3, 4][*]**    **Zhijiang Guo[5, 6][†]**
**Terry Yue Zhuo[7, 8]**    **Qianru Zhang[1][†]**    **Jie M. Zhang[9]**    **Heming Cui [1]**
**Siu-Ming Yiu[1]**    **Dong Huang[1, 4][†]**    **See-Kiong Ng[4]**    **Luu Anh Tuan[3]**

[1]HKU    [2]UCL    [3]NTU    [4]NUS    [5]HKUST (GZ)    [6]HKUST
[7]Monash University    [8]CSIRO's Data61    [9]KCL

## Abstract

Existing code generation benchmarks primarily evaluate functional correctness, with limited attention to code efficiency, and they are often restricted to a single language such as Python. To address this gap, we introduce EFFIBENCH-X, the first multi-language benchmark designed to measure the efficiency of LLM-generated code. EFFIBENCH-X supports Python, C++, Java, JavaScript, Ruby, and Golang. It comprises competitive programming tasks with human-expert solutions as efficiency baselines. Evaluating state-of-the-art LLMs on EFFIBENCH-X reveals that while models generate functionally correct code, they consistently underperform human experts in efficiency. Even the most efficient LLM-generated solutions (Qwen3-32B) achieve only around **62%** of human efficiency on average, with significant language-specific variations. LLMs show better efficiency in Python, Ruby, and JavaScript than in Java, C++, and Golang. For instance, DeepSeek-R1's Python code is significantly more efficient than its Java code. These results highlight the critical need for research into LLM optimization techniques to improve code efficiency across diverse languages. The dataset and evaluation infrastructure are publicly available at `https://github.com/EffiBench/EffiBench-X.git` and `https://huggingface.co/datasets/EffiBench/effibench-x`.

## 1   Introduction

Code efficiency is becoming a critical concern for LLM-generated code as these models are more widely adopted. While many LLMs and agent frameworks successfully produce correct solutions [16, 36, 77, 39, 30], their outputs often incur substantial resource overhead, compromising performance and feasibility in settings demanding high efficiency, such as mobile phones, embedded systems, and latency-sensitive cloud environments [48, 32, 66, 31, 33]. This underscores the need for benchmarks that measure not just correctness but also runtime efficiency [56, 66, 48].

Responding to this critical need, several benchmarks have recently emerged to measure the efficiency of LLM-generated code. EffiBench [32] and Mercury [23] use LeetCode problems and their solutions to evaluate Python code based on runtime and memory. Moving beyond LeetCode, EvalPerf [48] and ENAMEL [56] assess efficiency using subsets of existing benchmarks HumanEval [16] and MBPP [11], while PIE [62] benchmarks efficiency by compiling performance-improving edits across various CodeNet challenges. Despite these valuable contributions, a closer examination reveals limitations in current benchmarks, which hinder their effectiveness in accurately evaluating efficiency.

---

[*]Equal Contribution.
[†]Corresponding Authors.

39th Conference on Neural Information Processing Systems (NeurIPS 2025) Track on Datasets and Benchmarks.

Specifically, current code efficiency benchmarks exhibit three key limitations. First, they suffer from **Language Diversity** issues, primarily focusing on single-language measurement [32, 23, 66, 56, 48], often Python, despite its relatively small share (25%) of the overall programming landscape[†]. This narrow focus overlooks crucial language-specific factors like compiler optimizations and memory management prevalent in languages such as C++, Java, and Golang, necessitating a multi-language benchmark. Second, **Data Contamination** is a significant problem, as many benchmarks, including EffiBench and Mercury (using LeetCode) and ENAMEL and EvalPerf (using HumanEval and MBPP), rely on dated and widely circulated problem sets. These have often been "seen" by models during training [12, 40, 73, 22], leading to performance metrics that reflect memorization rather than genuine reasoning or optimization, thereby reducing their representativeness for evaluating performance on novel challenges. Third, current benchmarks are hampered by **Limited Complexity**, featuring tasks that are often straightforward and solvable without advanced algorithms. Benchmarks like HumanEval and MBPP, used by EvalPerf and ENAMEL, include simple prompts where models have already achieved high pass rates (e.g., >95% `pass@1` on HumanEval)[†]. Such trivial tasks fail to reveal critical performance differentials or require substantial computational resources, making them unsuitable for measuring efficiency and highlighting the need for benchmarks that incorporate more complex, computationally intensive problems to better assess an LLM's ability to handle large inputs or implement optimized solutions for real-world applications.

To address critical limitations in evaluating LLM code efficiency, we introduce EFFIBENCH-X, the first large-scale multi-language benchmark specifically designed for robust efficiency evaluation. EFFIBENCH-X evaluates efficiency across six diverse programming languages: Python, C++, Java, JavaScript, Ruby, and Golang, directly tackling the language diversity problem. It comprises recently released competitive programming tasks from various platforms, paired with canonical human expert solutions, effectively mitigating data contamination. By focusing on complex problems requiring advanced algorithms and data structures, EFFIBENCH-X overcomes the issue of limited complexity, providing a more accurate assessment of LLMs' efficiency in challenging scenarios. Furthermore, our comprehensive evaluation framework with high-resolution profiling ensures robust and reliable measurement of LLM-generated code efficiency within a controlled environment.

Leveraging EFFIBENCH-X and its comprehensive evaluation framework, we conducted a comprehensive study evaluating the performance of a wide range of LLMs against expert-written baselines. Our experiments reveal that while most LLMs are capable of generating functionally correct solutions, they consistently exhibit shortcomings in efficiency. Specifically, even the most efficient code produced by an LLM achieves only approximately **62%** of the runtime performance demonstrated by expert-crafted code. These findings highlight a significant gap and underscore the critical need for further research into optimization techniques. Such advancements are essential to empower LLMs to generate code that is not only correct but also highly efficient across diverse programming languages.

## 2  Related Work

Early efforts on code generation with LLMs gravitated toward assessing correctness and functionality, exemplified by HumanEval [16] and MBPP [11], which challenge models to produce valid code snippets from natural language docstrings. Building on this foundation, subsequent works have aimed to mitigate issues of limited diversity in tasks. For instance, HumanEval-X [76], MultiPLe [14], CodeScope [71] and MBXP [10] expand these benchmarks to multiple programming languages, while DS-1000 [42], ARCADE [74], and NumpyEval [75] target data science-oriented tasks. Beyond these, efforts like APIBench [55], BigCodeBench [77], BiasBench [34], and RepoBench [49] delve into broader software engineering subtasks, including library usage and repository-level code completion. While these benchmarks have been instrumental in gauging a model's correctness, they rarely address performance, potentially overlooking inefficiencies that hinder real-world adoption. Recognizing this significant limitation, a new wave of evaluations has emerged. As summarized in Table 1, while benchmarks such as EffiBench [32], EffiBench+ [72], Mercury [23], EvalPerf [48], ENAMEL [56], and PIE [62] specifically target performance metrics like runtime and memory usage, these valuable efforts commonly face limitations. Most are confined to single-language settings, predominantly Python. The multilingual benchmark CodeScope [71], for instance, provides only 30 tasks for its efficiency evaluation across four languages—Python 3, C, C#, and C++. Furthermore, many

---

[†]https://www.tiobe.com/tiobe-index/
[†]HumanEval Leaderboard: `https://paperswithcode.com/sota/code-generation-on-humaneval`.

Table 1: Comparison of EFFIBENCH-X to other code efficiency benchmarks. EFFIBENCH-X covers six programming languages, with one human-written solution for each language per task.

| Dataset | Tasks | Test Cases | Solutions | Metrics | Language | Source From |
|---|---|---|---|---|---|---|
| EvalPerf | 121 | 1/284.78 | 7.6 | $DPS/DPS_{norm}$ | Python | EvalPlus/HumanEval/MBPP |
| ENAMEL | 142 | 20 | 1 | Eff@k | Python | HumanEval |
| COFFE (Function) | 398 | 10.71 | 1 | Efficiency@k | Python | HumanEval/MBPP/APPS/CodeContests |
| COFFE (File) | 358 | 48.63 | 66.93 | Efficiency@k | Python | HumanEval/MBPP/APPS/CodeContests |
| PIE | 41 | 104 | 23.8 | %Opt/Speedup/%Correct | C++ | CodeNet |
| ECCO | 48 | 20.0 | 16.5 | Time/Memory | Python | CodeNet |
| Mercury | 256 | $+\infty$ | 18.4 | Pass/Beyond | Python | LeetCode |
| EffiBench | 1000 | 100 | 1 | ET/MU/TMU | Python | LeetCode |
| EffiBench+ | 160 | 100 | 10.7 | GET/ECC | Python | LeetCode |
| CodeScope (Optimization) | 30 | - | - | Opt@K/Time/Memory | Python3,C,C#,C++ | Codeforces |
| EFFIBENCH-X | 623 | 100 | 1 (6) | ET/MP/MI | Multiple | Competitions |

of these benchmarks rely on popular tasks (e.g., HumanEval and MBPP), which increases the risk of data contamination from LLM training data. Moreover, the often-simple nature of these tasks does not adequately expose nuanced performance differences critical in complex scenarios. To overcome these shortcomings, EFFIBENCH-X offers a large-scale, multi-language evaluation across *Python, C++, Java, JavaScript, Ruby, and Golang*. By utilizing over 600 diverse and recent competitive programming tasks, paired with expert-optimized solutions, EFFIBENCH-X allows for a more thorough and reliable assessment of modern LLMs.

## 3 EFFIBENCH-X

### 3.1 Efficiency-Critical Problem Collection

The foundation of EFFIBENCH-X lies in a curated collection of programming problems sourced from a wide array of competitive programming platforms, which are commonly used to evaluate human developers' abilities in writing efficient algorithms. Sourcing from multiple platforms ensures diversity in problem style, constraints, and required algorithmic techniques. These platforms include Aizu [4], AtCoder [9], CodeChef [17], Codeforces [18], and LeetCode [43]. Problems are categorized into two types based on their input/output handling requirements:

**(1) Functional Problems** require the implementation of a specific function or class, typically receiving input via parameters and returning output directly. LeetCode problems often fall into this category. The benchmark infrastructure handles I/O serialization and deserialization via test templates.

**(2) Standard I/O (stdio) Problems** require the implementation of a complete program that reads input from standard input (stdin) and writes output to standard output (stdout). This format is common on platforms like Codeforces.

Our two problem types allow EFFIBENCH-X to evaluate LLM capabilities in both library-like function generation and standalone program creation. A critical aspect of our collection process is *mitigating data contamination*, where LLMs might have encountered benchmark problems during their pre-training phase. To address this, we meticulously collect the original release date for each problem. We prioritize and filter for problems released after October 2023 [†], significantly reducing the likelihood that contemporary LLMs have been trained on them. This focus on recent problems ensures that EFFIBENCH-X primarily evaluates the models' generalization and reasoning capabilities rather than memorization. Furthermore, the competitive programming origin of these tasks inherently selects for problems demanding non-trivial algorithmic thinking and efficient data structure usage, addressing the limitation of benchmarks based on overly simplistic tasks. All models are evaluated under a unified instruction template and generation configurations (e.g., `temperature = 0`) to ensure fairness and consistency.

### 3.2 Canonical Solution Construction

To establish a reliable baseline for efficiency comparison, each problem in EFFIBENCH-X is paired with a few canonical solutions for each target language. These canonical solutions aim to represent high-quality, efficient code typically authored by human experts. We gather potential canonical

---

[†]October 2023 represents a common knowledge cutoff date for many contemporary LLMs, including GPT-4o and o1 models, reducing the likelihood that these problems appeared in their training data.

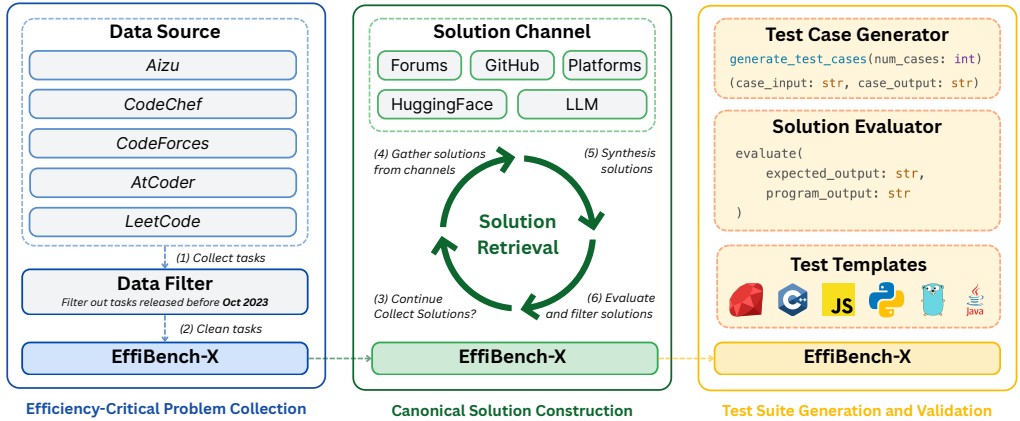

Figure 1: Overview of the construction pipeline. The process begins with collecting efficiency-critical problems from competitive programming platforms, followed by constructing canonical solutions from expert programmers, and then generating test suites with test case generators and solution evaluators.

solutions through multiple channels: (1) Publicly available solutions from platform discussion forums (e.g., LeetCode Discussion). (2) Publicly available accepted submissions retrieved via platform APIs. (3) Solutions curated from open-source repositories on GitHub and datasets on Hugging Face.

For functional problems, we strive to collect canonical solutions in all six target languages. This comprehensive collection is crucial for the subsequent validation of language-specific test templates. For stdio problems, collecting a canonical solution in at least one language is sufficient for validating the test case generator and solution evaluator. When canonical solutions for specific languages are missing, we employ a careful translation process using state-of-the-art LLMs (o3-mini). The key objective during translation is to preserve the original algorithm, logic, and time/space complexity of the source human-written solution. We explicitly instruct the LLM to perform a direct translation without introducing optimizations or degradations. This ensures that the translated solutions maintain the efficiency characteristics of the original expert code, serving as a fair baseline. All collected and translated solutions undergo a rigorous verification process. We submit them back to their original platforms whenever possible. Only solutions that are accepted (i.e., pass all internal tests for correctness and efficiency within the limits) are retained as canonical solutions in EFFIBENCH-X. Invalid or inefficient submissions are discarded. This step guarantees that our baseline represents demonstrably correct and performant code. We collect multiple accepted solutions per problem-language pair when available. Consistent with prior work [32, 56], we establish one expert baseline per metric: lowest runtime for ET, lowest peak memory for MP, and lowest memory integral for MI. Each baseline must pass platform acceptance checks and preserve the original algorithmic complexity. Our goal is an optimized gold standard rather than average human performance.

### 3.3 Test Suite Generation and Validation

A robust test suite [23, 32, 35, 56] is essential for reliably evaluating both correctness and efficiency. For each problem, we generate a comprehensive test suite comprising three components:

#### 3.3.1 Core Components

**Test Case Generator.** Instead of directly generating many test cases with LLMs, which consumes excessive tokens and is not scalable, we prompt LLMs to generate a test case generator. `generate_test_cases(num_cases: int, seed: int)` is a Python function responsible for producing a list of test cases (including `input` and `output`). The generator is designed to create diverse test cases covering various scenarios, including boundary values, typical inputs, stress tests (large inputs), edge cases, and potential performance traps, guided by the problem constraints, to sufficiently evaluate a program's efficiency. One canonical solution is provided for the generator to produce the expected output for each test input. By default, EFFIBENCH-X generates 100 test

cases per problem using a fixed seed for reproducibility. The serialization format is kept simple (e.g., comma-separated values, space-separated values) to ensure easy parsing across all target languages.

**Solution Evaluator.** `evaluate(expected_output: str, program_output: str)` is a Python function that determines the correctness of a solution's output for a given test case. Crucially, the evaluator first deserializes both the expected and program outputs using logic consistent with the test case generator. It then performs a logical comparison based on the problem's requirements, rather than a simple string comparison. This allows for flexibility in output formatting (e.g., handling different ordering in lists if permissible).

**Test Templates (Functional Problems Only).** For each target language, a code template is generated for functional problems. This template provides the necessary boilerplate code (e.g., main function, I/O handling) to create a runnable program. It includes a specific placeholder string, `==Code Submission==`, for injecting the functional solution (typically a function or method implementation). The template is responsible for reading the serialized input string from stdin, deserializing it into the appropriate data types expected by the solution function, calling the solution function, serializing the returned value, and printing the serialized result to stdout.

### 3.3.2 Component Validation

The test suites undergo an initial generation followed by a rigorous, multi-stage validation process to ensure their integrity. This comprehensive workflow guarantees their reliability, cross-language consistency, and accuracy, providing a solid basis for evaluating correctness and efficiency.

**Test Case Validation.** We execute the generated `generate_test_cases` function to produce 100 test cases. If the count is incorrect, the generator is deemed invalid, and the entire test suite for that problem is regenerated.

**Test Template Validation.** We check if the LLM-generated test templates for all six target languages. If any templates are missing, we prompt the LLM specifically to generate the missing ones based on the context of the existing components.

**Test Suite Validation.** For *functional problems*, we combine its canonical solution with its corresponding test template for each target language. This complete program is then executed against all 100 generated test cases. The program's stdout is captured and compared against the expected output using the `evaluate` function. If the canonical solution fails any test case for a given language, the test template for that language is considered potentially faulty. We attempt to automatically repair the template by providing the LLM (o3-mini) with the template code, the canonical solution, the failing test case, and the error message (or incorrect output). We allow up to 3 repair attempts per template. If repair fails, the problem is flagged for manual inspection. This process ensures that the test templates correctly interface with known-good solutions. For *stdio problems*, we execute the canonical solution (in at least one available language) against all generated test cases, using the `evaluate` function to verify the output. If the canonical solution passes all test cases, the test case generator and solution evaluator are considered valid for this problem.

## 3.4 Sandboxed Execution Environment

To ensure fair and accurate performance measurements, all solution code executions are performed within a controlled sandbox to provide a reproducible execution environment. Our approach builds upon the concepts of [65, 24], but incorporates substantial proprietary enhancements, including a high-resolution profiler, standard I/O support, and numerous performance improvements to meet the rigorous demands of our benchmark. This environment isolates executions from the host system and from each other, minimizing interference and variability. We leverage Docker containers as our sandboxing mechanism. Each execution runs within a dedicated container based on a pre-built image specific to the programming language (e.g., official Python, GCC, OpenJDK images), as detailed in Appendix C.9. Crucially, to mitigate interference from other processes and ensure consistent access to computational resources, we pin each worker's container to specific physical CPU cores using the `cpuset-cpus` Docker option. Our infrastructure detects the system's CPU topology (mapping logical cores to physical cores) and assigns containers to distinct physical cores, preventing multiple benchmark executions from contending for the same core resources simultaneously. This sandboxed setup provides a consistent and isolated environment, critical for obtaining reliable runtime and memory usage measurements.

### 3.5 High-Resolution Performance Profiling

Accurate efficiency assessment necessitates precise measurement of resource consumption. To this end, we developed a custom profiler that integrates with a sandboxed execution environment. This profiler captures high-resolution data by periodically sampling the execution time and memory usage of the solution. The sampling occurs at a high frequency (0.1 milliseconds, i.e., 10 kHz) to accurately capture peak resource usage and rapid fluctuations. The profiler also enforces specified memory limits, terminating processes that exceed them and logging Out-Of-Memory events. For each execution on a test case, the profiler outputs time-series data consisting of timestamps and corresponding memory usage readings, which forms the basis for the efficiency metrics (Section 4).

## 4 Evaluation Metrics

To quantify the efficiency of LLM-generated solutions relative to human-expert-written solutions, we follow existing works [32, 56, 48, 23, 62, 66] and define three key metrics:

**Execution Time (ET)** measures the runtime performance of an LLM-generated solution compared to the human-expert-written solution. For each problem $i$, let $T_i^{\text{H}}$ be the execution time of the human-expert-written solution required to pass all test cases for the task, and $T_i^{\text{L}}$ be the execution time of the LLM-generated solution required to pass all test cases for the task. If the LLM-generated solution for problem $i$ fails to pass all test cases or encounters a runtime error (e.g., timeout, crash) [23, 56], its ET score $s_i^T$ for that problem is defined as zero. Otherwise, the score is computed as the ratio $\frac{T_i^{\text{H}}}{T_i^{\text{L}}}$, which is then clipped to the interval $[0, 1]$:

$$s_i^T = \text{clip}\left(\frac{T_i^{\text{H}}}{T_i^{\text{L}}}, 0, 1\right)$$

This clipping ensures that an LLM-generated solution performing faster than the human-expert-written solution is considered equally efficient (score of 1) for this metric [23, 56, 48], preventing disproportionate influence from exceptionally fast outliers. The overall ET is then calculated as the average of these individual scores across all $N$ evaluated problems, expressed as a percentage:

$$\text{ET (\%)} = \left(\frac{1}{N} \sum_{i=1}^{N} s_i^T\right) \times 100\%$$

A higher ET percentage indicates that, on average, LLM-generated solutions achieve runtime performance closer to, or as good as, the human-expert-written solutions (where 100% signifies performance equivalent to or better than the human-expert-written solution).

**Memory Peak (MP)** indicates the minimum memory required for the system (e.g., mobile devices) to execute the code for the test cases. MP evaluates the memory peak of LLM-generated code relative to the human-expert-written solution. For each problem $i$, $M_i^{\text{H}}$ denotes the memory peak of the expert-written solution, and $M_i^{\text{L}}$ denotes that of the LLM-generated solution. Similar to ET, if the solution fails, its individual memory score $s_i^M$ for that problem is defined as zero. Otherwise, the score is computed as the ratio $\frac{M_i^{\text{H}}}{M_i^{\text{L}}}$, which is then clipped to $[0, 1]$:

$$s_i^M = \text{clip}\left(\frac{M_i^{\text{H}}}{M_i^{\text{L}}}, 0, 1\right)$$

Similar to ET, the overall MP can be expressed as a percentage:

$$\text{MP (\%)} = \left(\frac{1}{N} \sum_{i=1}^{N} s_i^M\right) \times 100\%$$

A higher MP percentage suggests that LLM-generated solutions, on average, exhibit peak memory footprints comparable to or as good as human-expert-written solutions.

**Memory Integral (MI)** measures the overall memory consumption throughout a solution's execution by comparing the area under the memory-time curve for LLM-generated solutions against human-expert-written ones. The memory integral $A$ for a single execution is defined as $A = \int_0^{T_{\text{total}}} M(t)\, dt$,

Table 2: Results averaged across six programming languages on EFFIBENCH-X.

| Model Name | Execution Time | Memory Peak | Memory Integral | Pass@1 |
|---|---|---|---|---|
| DeepSeek-V3-0324 | 40.46% | 51.52% | 39.38% | 53.29% |
| DeepSeek-R1 | 61.33% | 69.41% | 60.06% | 72.79% |
| Llama-4-Scout-17B-16E-Instruct | 23.16% | 28.09% | 22.61% | 28.44% |
| Llama-4-Maverick-17B-128E-Instruct | 16.28% | 36.47% | 15.52% | 37.32% |
| Qwen3-8B | 45.44% | 51.64% | 45.11% | 53.50% |
| Qwen3-14B | 59.75% | 60.79% | 58.58% | 63.30% |
| Qwen3-32B | **62.21%** | 67.26% | **61.48%** | 70.41% |
| Qwen2.5-Coder-7B-Instruct | 24.37% | 25.40% | 24.01% | 25.74% |
| Qwen2.5-Coder-14B-Instruct | 32.06% | 34.19% | 31.23% | 34.88% |
| Qwen2.5-Coder-32B-Instruct | 36.66% | 39.12% | 36.05% | 39.94% |
| QwQ-32B | 31.60% | 35.51% | 31.38% | 36.78% |
| Gemma-3-4B-It | 9.69% | 16.74% | 9.21% | 17.15% |
| Gemma-3-12B-It | 15.53% | 27.64% | 14.26% | 28.25% |
| Gemma-3-27B-It | 16.62% | 32.52% | 15.21% | 33.49% |
| Phi-4 | 28.42% | 29.81% | 27.34% | 30.60% |
| Phi-4-Reasoning | 48.37% | 48.83% | 46.98% | 50.54% |
| Phi-4-Reasoning-Plus | 36.62% | 38.22% | 35.86% | 39.27% |
| GPT-4o-mini | 16.96% | 35.12% | 16.19% | 36.06% |
| GPT-4o | 24.53% | 42.61% | 24.00% | 43.61% |
| Claude-3.5-Haiku | 36.33% | 44.06% | 35.07% | 45.24% |
| Claude-3.7-Sonnet | 47.79% | 54.60% | 46.98% | 56.23% |
| Gemini-2.0-Flash | 30.61% | 47.15% | 28.57% | 48.56% |
| Gemini-2.0-Flash-Lite | 27.86% | 38.61% | 26.28% | 39.89% |
| Gemini-2.0-Flash-Thinking | 38.82% | 55.56% | 36.83% | 57.38% |
| Gemini-2.5-Flash | 40.42% | 65.13% | 38.22% | 68.08% |
| Gemini-2.5-Pro | 47.82% | **75.60%** | 45.08% | **79.43%** |

where $M(t)$ is the memory usage at time $t$ over the solution's total execution time $T_{\text{total}}$. This integral is numerically approximated based on the high-resolution profiling data (Section 3.5). Let $A_i^{\text{H}}$ be the memory integral of the expert-written solution for problem $i$, and $A_i^{\text{L}}$ be that of the LLM-generated solution. Similar to ET, if the solution fails, its individual MI score $s_i^A$ for that problem is defined as zero. Otherwise, the score is computed as the ratio $\frac{A_i^{\text{H}}}{A_i^{\text{L}}}$, which is then clipped to $[0, 1]$:

$$s_i^A = \text{clip}\left(\frac{A_i^{\text{H}}}{A_i^{\text{L}}}, 0, 1\right)$$

This clipping ensures that solutions with a smaller memory integral (i.e., $A_i^{\text{L}} < A_i^{\text{H}}$, resulting in a ratio $\frac{A_i^{\text{H}}}{A_i^{\text{L}}} > 1$) than the human-expert solution are considered equivalently efficient (score of 1) for this metric, preventing disproportionate influence from exceptionally memory-frugal solutions. The overall MI can be expressed as a percentage:

$$\text{MI}\,(\%) \;=\; \left(\frac{1}{N}\sum_{i=1}^{N} s_i^A\right) \times 100\%$$

A higher MI percentage indicates that, on average, LLM-generated solutions exhibit overall memory consumption (integrated over time) comparable to or as good as human-expert solutions.

## 5  Evaluation

**Testbed:** Efficiency evaluations are conducted on AWS i7ie.metal-48xl instances. These instances feature 4th generation Intel Xeon Scalable (Sapphire Rapids) processors with 96 physical cores (192 vCPUs) and 384 GiB of RAM. All code execution occurs within our sandboxed environment, detailed in Section 3.4 and implemented using Docker containers, thereby ensuring reliable efficiency evaluations by isolating processes from host machine variations and preventing inter-task interference. Each code execution is constrained by a 10-second timeout and a 1024 MiB memory limit.

**Models and Metrics:** As detailed in Table 6, various LLMs are selected for evaluation. Open-source LLMs include DeepSeek-R1 [26] and DeepSeek-V3-0324 [47]; Llama-4-Scout-17B-16E-Instruct and

Table 3: Results on EFFIBENCH-X-C++ and EFFIBENCH-X-Python.

| Model Name | ET | MP | MI | Pass@1 | ET | MP | MI | Pass@1 |
|---|---|---|---|---|---|---|---|---|
| | C++ | | | | Python | | | |
| DeepSeek-V3-0324 | 39.02% | 52.54% | 32.23% | 54.41% | 48.02% | 53.22% | 48.59% | 55.38% |
| DeepSeek-R1 | 60.89% | 71.57% | 51.89% | 75.12% | 67.30% | 69.66% | 66.27% | 74.64% |
| Qwen3-14B | 61.22% | 63.84% | **57.03%** | 66.77% | 65.28% | 64.36% | 63.64% | 68.38% |
| Qwen3-32B | **63.89%** | 70.30% | 56.48% | 74.80% | **69.28%** | 70.94% | **68.34%** | 75.44% |
| Qwen2.5-Coder-14B-Instruct | 32.02% | 35.09% | 28.17% | 35.63% | 31.89% | 34.26% | 31.89% | 34.99% |
| Qwen2.5-Coder-32B-Instruct | 35.57% | 38.43% | 32.12% | 39.33% | 37.13% | 39.29% | 36.89% | 40.45% |
| QwQ-32B | 34.33% | 38.60% | 30.29% | 40.45% | 38.82% | 40.10% | 38.35% | 42.05% |
| Gemma-3-4B-It | 9.51% | 20.77% | 6.48% | 20.87% | 14.54% | 18.95% | 15.42% | 19.26% |
| Gemma-3-12B-It | 13.76% | 32.44% | 7.46% | 32.58% | 18.72% | 23.34% | 19.95% | 23.60% |
| Gemma-3-27B-It | 12.93% | 38.32% | 5.77% | 38.68% | 25.39% | 32.78% | 27.13% | 33.87% |
| Phi-4 | 28.60% | 30.56% | 24.08% | 31.46% | 30.14% | 31.71% | 29.63% | 32.74% |
| Phi-4-Reasoning | 42.63% | 42.77% | 36.59% | 45.10% | 50.86% | 50.76% | 49.92% | 53.45% |
| GPT-4o-mini | 10.28% | 35.32% | 6.07% | 36.12% | 27.12% | 36.21% | 28.26% | 37.40% |
| GPT-4o | 12.62% | 37.27% | 7.58% | 37.88% | 38.05% | 48.13% | 40.24% | 48.96% |
| Claude-3.5-Haiku | 35.17% | 45.84% | 31.00% | 47.03% | 34.46% | 40.41% | 34.86% | 41.41% |
| Claude-3.7-Sonnet | 45.45% | 55.26% | 40.68% | 56.98% | 51.90% | 55.88% | 52.03% | 57.78% |
| Gemini-2.5-Flash | 35.84% | 68.26% | 23.52% | 71.91% | 56.08% | 66.14% | 58.30% | 69.82% |
| Gemini-2.5-Pro | 43.33% | **75.51%** | 26.60% | **81.06%** | 66.12% | **75.08%** | 67.72% | **79.94%** |

Llama-4-Maverick-17B-128E-Instruct [51]; Qwen3 (8B, 14B, 32B) [57], Qwen2.5-Coder-Instruct (14B, 32B) [37], and QwQ-32B [58]; Gemma-3-It (4B, 12B, 27B) [63]; and Phi-4, Phi-4-Reasoning, and Phi-4-Reasoning-Plus [1]. Proprietary LLMs include GPT-4o and GPT-4o-mini [38]; Claude-3.5-Haiku and Claude-3.7-Sonnet [8, 7]; Gemini-2.0 (Flash, Flash-Lite, Flash-Thinking) and Gemini-2.5 (Flash, Pro) [6]. For evaluation metrics, we use Execution Time (ET), Memory Peak (MP), Memory Integral (MI), and Pass@1 [15] to measure the correctness of the generated code.

## 5.1 Main Results

**Open-source LLMs:** As shown in Table 2, among the evaluated open-source models, DeepSeek-R1 demonstrates exceptional performance, achieving the highest scores among open-source models with an ET of 61.33%, MP of 69.41%, MI of 60.06%, and a Pass@1 rate of 72.79%. The Qwen3 series also shows strong performance, particularly Qwen3-32B, which slightly outperforms DeepSeek-R1 in execution time efficiency with an ET of 62.21% and MI of 61.48%, while achieving a solid Pass@1 of 70.41%. The Phi-4 family shows interesting variations, with Phi-4-Reasoning achieving notably better performance (ET 48.37%, Pass@1 50.54%) compared to the base Phi-4 model (ET 28.42%, Pass@1 30.60%). Other models like the Llama-4 series generally exhibit lower performance in both correctness and efficiency metrics, with Llama-4-Maverick-17B-128E-Instruct scoring an ET of just 16.28% and Pass@1 of 37.32%. This highlights a significant variance in capabilities among currently available open-source LLMs for generating efficient and correct code.

**Proprietary LLMs:** The proprietary LLMs generally show strong performance. Gemini-2.5-Pro emerges as the top-performing model overall, with the highest MP of 75.60% and Pass@1 rate of 79.43%, though its ET (47.82%) and MI (45.08%) fall short of the best open-source models. Gemini-2.5-Flash also delivers robust results (ET 40.42%, MP 65.13%, MI 38.22%, Pass@1 68.08%). Models like Claude-3.7-Sonnet (ET 47.79%, Pass@1 56.23%) and GPT-4o (ET 24.53%, Pass@1 43.61%) offer competitive, albeit lower, performance levels. This suggests that while leading proprietary models often set the benchmark for correctness and memory peak efficiency, some open-source models are now achieving superior execution time efficiency.

**Impact of Model Size:** Model size within a family generally correlates positively with performance. Examining the Qwen3 series, Qwen3-8B achieves an ET of 45.44% and Pass@1 of 53.50%. This scales up with Qwen3-14B (ET 59.75%, Pass@1 63.30%), and further with Qwen3-32B (ET 62.21%, Pass@1 70.41%). A similar trend is observed in the Qwen2.5-Coder series, with the 32B variant (ET 36.66%, Pass@1 39.94%) outperforming the 14B counterpart (ET 32.06%, Pass@1 34.88%). The Gemma-3 series follows this pattern as well, with performance increasing from Gemma-3-4B-It (ET 9.69%, Pass@1 17.15%) to Gemma-3-27B-It (ET 16.62%, Pass@1 33.49%). This indicates that larger models tend to have better capabilities for generating both correct and more efficient code.

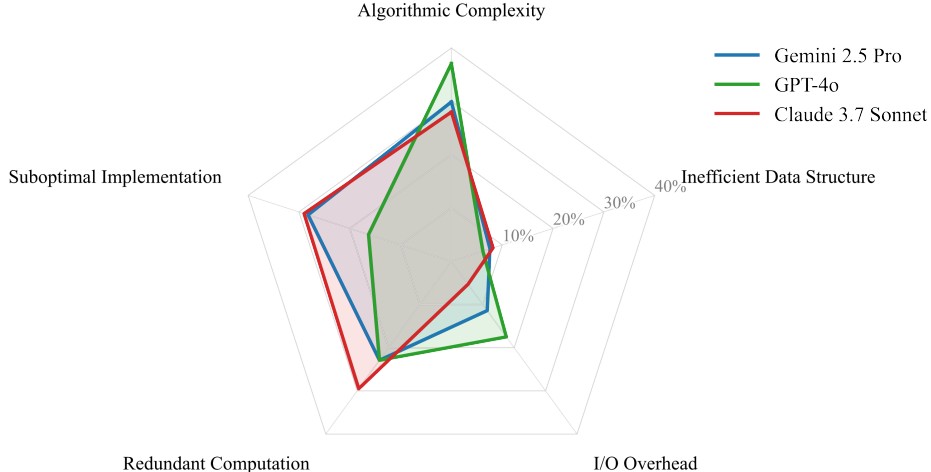

Figure 2: Distribution of inefficiency causes across representative proprietary models.

Table 4: Inefficiency Analysis by Category (higher = more frequent cause).

| Inefficiency Category | Gemini-2.5-Pro | GPT-4o | Claude-3.7-Sonnet |
|---|---|---|---|
| Algorithmic Complexity | **29.9%** | **37.1%** | 28.0% |
| Suboptimal Implementation | 28.2% | 16.3% | **29.0%** |
| Redundant Computation | 22.9% | 22.9% | **29.5%** |
| I/O Overhead | 11.4% | 17.5% | 5.3% |
| Inefficient Data Structure | 7.6% | 6.2% | 8.2% |

**Impact of Model Specialization:** Newer versions and specialized variants of models often demonstrate considerable performance improvements. The Gemini family illustrates this clearly: Gemini-2.0-Flash shows an ET of 30.61% and Pass@1 of 48.56%, while the specialized Gemini-2.0-Flash-Thinking variant improves upon this with an ET of 38.82% and Pass@1 of 57.38%. Further advancements are seen with Gemini-2.5-Flash (ET 40.42%, Pass@1 68.08%), and Gemini-2.5-Pro leads in correctness (Pass@1 79.43%). Similar benefits can be observed in the Phi-4 series, where the reasoning-enhanced variant (ET 48.37%, Pass@1 50.54%) significantly outperforms the base model (ET 28.42%, Pass@1 30.60%). This trend highlights the rapid evolution in LLM capabilities and the benefits of targeted model enhancements.

Overall, while several state-of-the-art LLMs can generate functionally correct code at high rates, there remains a significant gap in achieving efficiency comparable to human expert solutions. Even the best-performing models achieve around 60-62% relative execution time on average, indicating substantial room for improvement in generating truly optimized code.

## 5.2 Sources of Inefficiency and Task-Type Analysis

**What makes model code slow?** Across models, the dominant failure mode is *Algorithmic Complexity* rather than micro-optimizations: GPT-4o shows the highest share of complexity-driven inefficiency (37.1%), while Claude 3.7 Sonnet is most affected by *Suboptimal Implementation* (29.0%) and *Redundant Computation* (29.5%); Gemini 2.5 Pro is also primarily limited by algorithmic choices (29.9%) (Figure 2 and Table 4). This indicates that improving algorithm selection and pruning strategies would yield the largest efficiency gains, beyond language-specific I/O tweaks or data-structure substitutions.

## 5.3 Comparison on Different Language Subsets

We verify the consistency of model performance across programming languages by analyzing results on the C++ and Python subsets (Table 3). This analysis confirms that the relative ranking of models observed in the end-to-end results remains largely consistent across individual language subsets, despite variations in absolute performance values. In both C++ and Python, the top-performing models

Table 5: DeepSeek-R1 and Claude-3.7-Sonnet on EFFIBENCH-X across different languages.

| Language | ET | MP | MI | Pass@1 | ET | MP | MI | Pass@1 |
|---|---|---|---|---|---|---|---|---|
| | | DeepSeek-R1 | | | | Claude-3.7-Sonnet | | |
| JavaScript | 63.34% | 69.19% | 63.03% | 71.43% | 50.15% | **57.33%** | 49.48% | **58.59%** |
| Ruby | 64.01% | 66.11% | 63.65% | 69.02% | 49.44% | 52.48% | 49.59% | 53.61% |
| Python | **67.30%** | 69.66% | **66.27%** | 74.64% | **51.90%** | 55.88% | **52.03%** | 57.78% |
| Java | 52.23% | 69.36% | 54.98% | 73.19% | 46.33% | 55.11% | 46.41% | 56.66% |
| C++ | 60.89% | **71.57%** | 51.89% | **75.12%** | 45.45% | 55.26% | 40.68% | 56.98% |
| Go | 60.24% | 70.57% | 60.57% | 73.35% | 43.48% | 51.52% | 43.67% | 53.77% |

maintain their leadership positions: Qwen3-32B and DeepSeek-R1 excel in execution time efficiency across both languages (C++: 63.89% and 60.89%; Python: 69.28% and 67.30%, respectively), while Gemini-2.5-Pro consistently leads in Pass@1 (C++: 81.06%; Python: 79.94%) and memory peak metrics (C++: 75.51%; Python: 75.08%). Similarly, mid-tier models like Claude-3.7-Sonnet and Phi-4-Reasoning maintain their relative positions, as do lower-performing models like the Gemma-3 series. While absolute performance tends to be higher on Python than C++ across most models, these variations affect all models similarly, preserving their relative standings. This cross-language consistency validates the robustness of EFFIBENCH-X, confirming that a model's general code efficiency capabilities transfer across diverse programming contexts.

## 5.4 Language-Specific Performance

We provide the evaluation results of DeepSeek-R1 and Claude-3.7-Sonnet on EFFIBENCH-X across different programming languages in Table 5. Both models demonstrate strong functional correctness across languages, with DeepSeek-R1 achieving Pass@1 rates from 69.02% (Ruby) to 75.12% (C++), and Claude-3.7-Sonnet ranging from 53.61% (Ruby) to 58.59% (JavaScript). However, we observe significant variations in efficiency metrics across language types. Dynamically-typed languages (Python, Ruby, JavaScript) consistently show higher execution time efficiency, with Python leading at 67.30% ET for DeepSeek-R1 and 51.90% ET for Claude-3.7-Sonnet, while statically-typed languages like Java show comparatively lower ET scores. This language-based efficiency gap persists across both models despite strong functional correctness, suggesting that LLMs may have developed better optimization strategies for widely used scripting languages. The contrast is particularly notable with C++, where DeepSeek-R1 achieves its highest Pass@1 (75.12%) and strong memory performance (MP: 71.57%), yet shows only 60.89% execution time efficiency compared to Python's 67.30%. These findings indicate that while current LLMs can generate syntactically correct and functionally working code across languages, their ability to produce optimized code that matches human efficiency varies substantially by language type, highlighting an area for future research.

## 6 Conclusion

We propose EFFIBENCH-X, the first large-scale multi-language benchmark specifically designed for robust efficiency evaluation across six programming languages. Our comprehensive assessment of 26 SOTA LLMs reveals significant efficiency gaps between LLM-generated and human expert code, with even the best-performing model (Qwen3-32B) achieving only about 62% of human-level efficiency on average. Performance varies considerably by language. These findings underscore the critical need to enhance LLMs' optimization capabilities across diverse languages, particularly where correctness and efficiency are vital. EFFIBENCH-X serves as a resource for future research aimed at improving multilingual code generation.

## Acknowledgments

The work is supported in part by National Key R&D Program of China (2022ZD0160201), HK RGC RIF (R7030-22), HK RGC GRF (ref No.: 17208223 & 17204424), and the HKU-CAS Joint Laboratory for Intelligent System Software. This research is also supported by A*STAR, CISCO Systems (USA) Pte. Ltd and National University of Singapore under its Cisco-NUS Accelerated Digital Economy Corporate Laboratory (Award I21001E0002), and DSO grant DSOCL23216.

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

# A   Limitations

Despite the significant contributions of EFFIBENCH-X in establishing a multi-language benchmark for code efficiency using novel and complex competitive programming tasks, our study and benchmark have several limitations that warrant discussion. Firstly, while competitive programming problems effectively address data contamination and complexity, this focus primarily evaluates algorithmic and data structure efficiency. It may not fully encompass the spectrum of efficiency considerations critical in other software development domains, such as low-level system programming, optimizing for specific hardware architectures, or managing large-scale data processing pipelines, thus potentially limiting the generalizability of findings to all efficiency-critical code generation contexts.

To address this limitation, future work will expand EFFIBENCH-X to incorporate system-level efficiency dimensions. Specifically, our planned extensions include: (1) introducing new benchmarks from HPC and I/O-intensive domains to test vectorization, cache locality, and asynchronous I/O; (2) capturing granular system metrics such as cache misses and I/O throughput using profiling tools; and (3) developing expert-written baselines to establish a stronger performance reference. These directions aim to extend EFFIBENCH-X toward a more holistic representation of efficiency across both algorithmic and systems levels.

Our current analysis primarily quantifies the efficiency gap between LLM-generated and expert code; it does not delve deeply into the root causes of LLM inefficiency at a granular code level (e.g., identifying specific code constructs or algorithmic choices that are suboptimal), which is useful for providing targeted feedback for model improvement. Finally, conducting extensive efficiency evaluations across multiple languages, numerous models, and complex problems with high-resolution profiling is computationally demanding, which could present a practical barrier for researchers with limited resources wishing to replicate or significantly extend this evaluation on a large scale.

# B   Broader Impacts

**Positive Societal Impacts**   The development and adoption of benchmarks like EFFIBENCH-X are crucial for driving progress in LLM-generated code efficiency. This progress promises significant positive societal impacts, including reduced energy consumption and lower operational costs for computing resources at scale, as inefficient code requires more energy and infrastructure. Furthermore, enabling LLMs to generate more efficient code is critical for deploying these models and the applications they help create in resource-constrained environments such as mobile devices and embedded systems, as well as in latency-sensitive cloud applications where performance is paramount. Benchmarks like ours also provide the research community with a clearer understanding of current LLM capabilities and limitations regarding code efficiency across diverse languages, fostering targeted research towards more effective and beneficial LLM-driven development workflows.

**Negative Societal Impacts**   However, improvements in LLM capabilities, including enhanced code efficiency facilitated by benchmarks like EFFIBENCH-X, also carry potential negative societal impacts. As LLMs become more adept at generating highly optimized code, there is a risk that these powerful capabilities could be exploited for malicious purposes, such as creating more efficient and evasive malware or facilitating sophisticated cyberattacks. This potential advancement in LLM-generated harmful code could inadvertently lower the technical barrier for malicious actors. Additionally, while efficiency is a goal, relying heavily on LLM-generated code necessitates robust validation processes; even efficient code could contain subtle bugs or security vulnerabilities that might be overlooked, potentially introducing new risks at scale. Addressing these concerns requires responsible development and deployment practices for LLMs, coupled with continued research into security, interpretability, and comprehensive validation techniques.

Table 6: Model list and URLs.

| Model Name | URL |
|---|---|
| **Open Source Models** | |
| DeepSeek-V3-0324 | `https://huggingface.co/deepseek-ai/DeepSeek-V3-0324` |
| DeepSeek-R1 | `https://huggingface.co/deepseek-ai/DeepSeek-R1` |
| Llama-4-Scout-17B-16E-Instruct | `https://huggingface.co/meta-llama/Llama-4-Scout-17B-16E-Instruct` |
| Llama-4-Maverick-17B-128E-Instruct | `https://huggingface.co/meta-llama/Llama-4-Maverick-17B-128E-Instruct` |
| Qwen3-8B | `https://huggingface.co/Qwen/Qwen3-8B` |
| Qwen3-14B | `https://huggingface.co/Qwen/Qwen3-14B` |
| Qwen3-32B | `https://huggingface.co/Qwen/Qwen3-32B` |
| Qwen2.5-Coder-7B-Instruct | `https://huggingface.co/Qwen/Qwen2.5-Coder-7B-Instruct` |
| Qwen2.5-Coder-14B-Instruct | `https://huggingface.co/Qwen/Qwen2.5-Coder-14B-Instruct` |
| Qwen2.5-Coder-32B-Instruct | `https://huggingface.co/Qwen/Qwen2.5-Coder-32B-Instruct` |
| QwQ-32B | `https://huggingface.co/Qwen/QwQ-32B` |
| Gemma-3-4B-It | `https://huggingface.co/google/gemma-3-4b-it` |
| Gemma-3-12B-It | `https://huggingface.co/google/gemma-3-12b-it` |
| Gemma-3-27B-It | `https://huggingface.co/google/gemma-3-27b-it` |
| Phi-4 | `https://huggingface.co/microsoft/phi-4` |
| Phi-4-Reasoning | `https://huggingface.co/microsoft/phi-4-reasoning` |
| Phi-4-Reasoning-Plus | `https://huggingface.co/microsoft/phi-4-reasoning-plus` |
| **Proprietary Models** | |
| GPT-4o-mini | `https://platform.openai.com/docs/models/gpt-4o-mini` |
| GPT-4o | `https://platform.openai.com/docs/models/gpt-4o` |
| Claude-3.5-Haiku | `https://www.anthropic.com/claude/haiku` |
| Claude-3.7-Sonnet | `https://www.anthropic.com/claude/sonnet` |
| Gemini-2.0-Flash | `https://cloud.google.com/vertex-ai/generative-ai/docs/models/gemini/2-0-flash` |
| Gemini-2.0-Flash-Lite | `https://cloud.google.com/vertex-ai/generative-ai/docs/models/gemini/2-0-flash-lite` |
| Gemini-2.0-Flash-Thinking | `https://cloud.google.com/vertex-ai/generative-ai/docs/models/gemini/2-0-flash` |
| Gemini-2.5-Flash | `https://cloud.google.com/vertex-ai/generative-ai/docs/models/gemini/2-5-flash` |
| Gemini-2.5-Pro | `https://cloud.google.com/vertex-ai/generative-ai/docs/models/gemini/2-5-pro` |

# C  Technical Appendices and Supplementary Material

## C.1  Model List

## C.2  Reliability of Efficiency Measurement

To assess the reliability of our efficiency metrics and ensure robustness against potential execution variability, we conducted multiple runs of our evaluation pipeline. Table 7 presents the results of executing the code generated by DeepSeek-R1 and Claude-3.7-Sonnet three times, reporting the mean, minimum, and maximum values for each efficiency metric. The results demonstrate high consistency across repeated executions. For execution time (ET), which might be expected to show the most variability due to system load fluctuations, we observe relatively small variations. DeepSeek-R1 shows an overall mean ET of 62.45% with a range from 61.33% to 63.55% (a variation of approximately ±1.7%). Similarly, Claude-3.7-Sonnet achieves a mean ET of 48.63% with a range from 47.79% to 49.31% (a variation of about ±1.5 percentage points). This narrow range confirms the reliability of our execution time measurements. Memory metrics show even greater stability across runs. For memory peak (MP), both models demonstrate extremely consistent results, with variations of less than 0.1 percentage points (DeepSeek-R1: 69.42% with range 69.41%-69.44%; Claude-3.7-Sonnet: 54.63% with range 54.60%-54.65%). Memory integral (MI) shows slightly more variation but remains highly stable (DeepSeek-R1: 61.15% with range 60.06%-62.19%; Claude-3.7-Sonnet: 47.78% with range 46.98%-48.40%). Language-specific results follow similar patterns of consistency. For instance, Python shows the highest stability with ET ranges of 67.30%-67.67% for DeepSeek-R1 and 51.90%-52.06% for Claude-3.7-Sonnet. Java exhibits slightly wider variations (DeepSeek-R1: 52.23%-57.56%; Claude-3.7-Sonnet: 46.33%-49.01%), potentially reflecting the additional variability introduced by JVM optimization and garbage collection processes. However, even these wider ranges remain relatively narrow, preserving the relative performance relationships between models.

These results validate the robustness of our efficiency measurement approach. The sandboxed execution environment, combined with our methodology of executing code multiple times, effectively controls for transient system variations while capturing meaningful differences in code efficiency. The consistent results across repeated runs confirm that the efficiency metrics reported throughout this paper reliably represent the true performance characteristics of LLM-generated code, providing a solid foundation for comparative analysis across models and languages.

Table 7: Robustness evaluation of code efficiency metrics through triple execution of DeepSeek-R1 and Claude-3.7-Sonnet generated solutions. Results are reported in Mean (min, max) format to demonstrate consistency across multiple runs.

| Model Name | ET | MP | MI | Pass@1 |
|---|---|---|---|---|
| All | | | | |
| DeepSeek-R1 | 62.45% (61.33%, 63.55%) | 69.42% (69.41%, 69.44%) | 61.15% (60.06%, 62.19%) | 72.79% |
| Claude-3.7-Sonnet | 48.63% (47.79%, 49.31%) | 54.63% (54.60%, 54.65%) | 47.78% (46.98%, 48.40%) | 56.21% |
| C++ | | | | |
| DeepSeek-R1 | 61.79% (60.89%, 62.77%) | 71.54% (71.44%, 71.61%) | 52.99% (51.89%, 54.10%) | 75.12% |
| Claude-3.7-Sonnet | 46.36% (45.45%, 47.18%) | 55.12% (55.03%, 55.26%) | 41.58% (40.68%, 42.05%) | 56.82% |
| Java | | | | |
| DeepSeek-R1 | 55.01% (52.23%, 57.56%) | 69.34% (69.31%, 69.36%) | 57.54% (54.98%, 59.82%) | 73.19% |
| Claude-3.7-Sonnet | 47.95% (46.33%, 49.01%) | 55.09% (55.06%, 55.11%) | 47.92% (46.41%, 49.04%) | 56.66% |
| JavaScript | | | | |
| DeepSeek-R1 | 64.09% (63.34%, 64.84%) | 69.20% (69.19%, 69.21%) | 63.71% (63.03%, 64.38%) | 71.43% |
| Claude-3.7-Sonnet | 50.86% (50.15%, 51.44%) | 57.31% (57.30%, 57.33%) | 50.19% (49.48%, 50.76%) | 58.59% |
| Ruby | | | | |
| DeepSeek-R1 | 64.14% (64.01%, 64.27%) | 66.14% (66.11%, 66.16%) | 63.78% (63.65%, 63.91%) | 69.02% |
| Claude-3.7-Sonnet | 49.61% (49.44%, 49.76%) | 52.48% (52.48%, 52.49%) | 49.73% (49.59%, 49.86%) | 53.61% |
| Golang | | | | |
| DeepSeek-R1 | 62.20% (60.24%, 64.16%) | 70.65% (70.57%, 70.71%) | 62.58% (60.57%, 64.59%) | 73.35% |
| Claude-3.7-Sonnet | 45.01% (43.48%, 46.41%) | 51.89% (51.52%, 52.11%) | 45.23% (43.67%, 46.66%) | 53.77% |
| Python | | | | |
| DeepSeek-R1 | 67.49% (67.30%, 67.67%) | 69.66% (69.66%, 69.67%) | 66.32% (66.27%, 66.36%) | 74.64% |
| Claude-3.7-Sonnet | 51.99% (51.90%, 52.06%) | 55.89% (55.88%, 55.90%) | 52.04% (52.00%, 52.09%) | 57.78% |

Table 8: Success rate of test-case generator.

| Attempts | # of Generators | Ratio (%) | Cumulative Ratio (%) |
|---|---|---|---|
| 1 | 510 | 81.86 | 81.86 |
| 2 | 71 | 11.40 | 93.26 |
| 3 | 20 | 3.21 | 96.47 |
| 4 | 4 | 0.64 | 97.11 |
| 5 | 3 | 0.48 | 97.59 |
| >5 | 15 | 2.41 | 100.00 |

## C.3 Test-Case Generator Metrics

To evaluate the reliability and comprehensiveness of our test-case generator, we measured both its generation success rate and code coverage across all tasks in EFFIBENCH-X. As shown in Table 8, the generator achieves a high success rate, successfully producing valid test cases on the first attempt for 81.86% of tasks, and reaching a cumulative success rate of 96.47% within three automated retries. Only a small fraction (2.41%) required more than five attempts, demonstrating the robustness of our generation process.

Furthermore, the generated tests exhibit full coverage when executed against the canonical solutions. Table 9 reports that our generated tests achieve 100% line and branch coverage, confirming that they comprehensively validate functional correctness and efficiency for each task.

## C.4 Clipped Efficiency Metrics

To further validate the robustness of our efficiency evaluation, we re-analyzed performance using *unclipped* execution time (ET) ratios. As shown in Table 10, models such as Claude-3.7-Sonnet produce faster-than-expert code in over 20% of cases, while maintaining the same relative model ranking observed under the clipped ET metric. These results confirm that our main conclusions are

Table 9: Coverage analysis of generated tests.

| Metric | Results (%) |
|---|---|
| Line Coverage | 100.00 |
| Branch Coverage | 100.00 |

Table 10: Unclipped median ET ratio and proportion of tasks with faster-than-expert performance (ET > 1.0).

| Model | Median Unclipped ET Ratio | % of Tasks with ET > 1.0 |
|---|---|---|
| GPT-4o-mini | 0.77 | 1.61 |
| GPT-4o | 0.82 | 7.54 |
| Claude-3.5-Haiku | 0.91 | 9.15 |
| Gemini-2.5-Flash | 0.87 | 13.80 |
| Gemini-2.5-Pro | 0.91 | 19.26 |
| Claude-3.7-Sonnet | 0.97 | 20.71 |

not artifacts of the clipping threshold but reflect genuine model differences in efficiency-oriented code generation.

To better understand the sources of these gains, we analyzed a subset of faster-than-expert solutions and categorized their optimization strategies (Table 11). Gemini-2.5-Pro and Claude-3.7-Sonnet primarily apply *implementation-level optimizations*—for example, achieving a $3.33\times$ speedup by avoiding string conversions inside loops—whereas GPT-4o more often exhibits *algorithmic or structural improvements*, such as reducing asymptotic complexity (e.g., $O(N^2)$ to $O(N \log N)$) or reformulating the problem entirely. This diversity of optimization behaviors highlights distinct reasoning pathways across model families and underscores the analytical depth of the evaluation framework.

### C.5 Dataset Structure and Composition

To enhance the transparency and reproducibility of EFFIBENCH-X, we provide detailed documentation of its structure and composition in Tables 12 and 13. Table 12 presents the dataset card of EFFIBENCH-X, describing the key fields and their definitions, which collectively define how each programming problem is represented and utilized during evaluation. Table 13 summarizes the distribution of problems across the five major competitive programming websites from which the dataset was constructed. This diversity ensures broad coverage across problem types, complexities, and constraints, contributing to a robust and reliable benchmark for evaluating code efficiency generation.

### C.6 Additional Language Subsets

Tables 14, 15, 16, and 17 present detailed evaluation results for the Java, JavaScript, Ruby, and Go subsets of EFFIBENCH-X. These findings reinforce the performance patterns observed in our primary analysis while revealing language-specific insights. For JavaScript, Qwen3-32B leads open-source models with 70.29% ET and 69.87% MI, slightly outperforming DeepSeek-R1, while Gemini-2.5-Pro achieves the highest Pass@1 (80.90%) and MP (78.22%). In Java, Qwen3-14B unexpectedly

Table 11: Categorization of optimization strategies observed in faster-than-expert code.

| Optimization Category | Gemini-2.5-Pro | Claude-3.7-Sonnet | GPT-4o |
|---|---|---|---|
| Implementation-Level Optimization | **55.83** | **62.02** | 13.79 |
| Algorithmic Complexity Optimization | 26.67 | 20.93 | 37.93 |
| Problem Reformulation | 7.50 | 6.98 | **41.38** |
| Advanced Data Structure Usage | 5.00 | 3.88 | 0.00 |
| Pruning and Heuristic Optimization | 5.00 | 6.20 | 6.90 |

Table 12: Dataset Card of EffiBench-X

| Field Name | Definition |
| --- | --- |
| id | Problem index in the corresponding source. |
| title | Task name. |
| title_slug | URL-friendly slug of the task name, used for referencing. |
| description | Task description. |
| description_md | Task description in markdown format. |
| source | The source platform from which the problem was collected. |
| url | URL to the original problem. |
| type | Problem type: functional or I/O. |
| starter_code | Starter code provided in multiple programming languages (C++, Java, Python, JavaScript, Go, Ruby). |
| solutions | Canonical solutions for each language. For each language, runtime represents the solution optimized for minimal runtime (with reasonably low memory), and memory represents the solution optimized for minimal memory (with reasonably low runtime). |
| test_case_generator | The test case generator used to produce evaluation test cases. |
| generated_tests | Tests generated by the test_case_generator, which are used in our experiments. |
| test_runner | Test templates for functional problems. For each target language, a runnable code template is generated to handle input/output serialization and function invocation. |

Table 13: Task distribution by source websites in EffiBench-X

| Website | Aizu | CodeChef | Codeforces | AtCoder | LeetCode | Total |
| --- | --- | --- | --- | --- | --- | --- |
| Tasks | 33 | 93 | 10 | 149 | 338 | 623 |

achieves the highest efficiency (62.32% ET, 62.02% MI) among all models, outperforming larger variants, while Gemini-2.5-Pro leads in correctness (80.42% Pass@1). For Ruby, Gemini-2.5-Pro demonstrates the best overall performance (65.66% ET, 74.32% Pass@1), with DeepSeek-R1 leading open-source models (64.01% ET, 69.02% Pass@1). In Go, Qwen3-32B shows the highest efficiency among open-source models (62.13% ET, 62.60% MI), while Gemini-2.5-Pro achieves the highest correctness (79.94% Pass@1) despite lower execution efficiency (31.50% ET). Several patterns emerge consistently across languages: model rankings remain relatively stable despite variations in absolute scores; dynamically-typed languages generally show higher efficiency scores than statically-typed languages; memory efficiency and correctness appear more closely correlated than execution time efficiency and correctness; and reasoning-enhanced models consistently outperform their base counterparts. These additional evaluations validate EffiBench-X's robustness as a comprehensive benchmark for assessing code efficiency generation across diverse programming contexts.

## C.7 Problem Types

Tables 18 and 19 present detailed evaluation results across two problem categories in our benchmark: functional problems and standard I/O problems. For functional problems, DeepSeek-R1 achieves the highest execution efficiency among open-source models (74.53% ET, 73.26% MI), followed closely by Qwen3-32B (73.14% ET, 72.85% MI) and Qwen3-14B (72.92% ET, 72.37% MI). Gemini-2.5-Pro excels in correctness and memory peak efficiency (90.04% MP, 92.50% Pass@1) but shows lower execution efficiency (57.41% ET) than the top open-source alternatives. Phi-4-Reasoning delivers strong performance (56.73% ET, 58.28% Pass@1), significantly outperforming its base model Phi-4 (30.54% ET, 32.79% Pass@1).

For standard I/O problems, performance drops substantially across all models. Qwen3-32B leads in execution efficiency (49.25% ET, 48.00% MI), followed by DeepSeek-R1 (45.68% ET, 44.42% MI) and Qwen3-14B (44.12% ET, 42.22% MI). Gemini-2.5-Pro maintains its leadership in correctness and memory metrics (58.47% MP, 63.92% Pass@1), though with much lower absolute scores than on

Table 14: Evaluation of code generated by various LLMs on EFFIBENCH-X-Java.

| Model Name ET | MP | MI | Pass@1 | |
|---|---|---|---|---|
| DeepSeek-V3-0324 | 34.46% | 52.11% | 34.35% | 53.93% |
| DeepSeek-R1 | 52.23% | 69.36% | 54.98% | 73.19% |
| Llama-4-Scout-17B-16E-Instruct | 20.92% | 26.16% | 20.46% | 26.48% |
| Llama-4-Maverick-17B-128E-Instruct | 11.24% | 38.80% | 8.45% | 39.81% |
| Qwen3-8B | 32.63% | 53.83% | 37.54% | 56.18% |
| Qwen3-14B | **62.32%** | 65.04% | **62.02%** | 68.70% |
| Qwen3-32B | 52.18% | 68.92% | 56.32% | 72.71% |
| Qwen2.5-Coder-7B-Instruct | 25.14% | 25.96% | 24.99% | 26.32% |
| Qwen2.5-Coder-14B-Instruct | 33.76% | 34.24% | 32.88% | 35.47% |
| Qwen2.5-Coder-32B-Instruct | 38.83% | 40.64% | 39.19% | 41.41% |
| QwQ-32B | 23.90% | 36.84% | 27.17% | 38.20% |
| Gemma-3-4B-It | 6.01% | 12.84% | 4.89% | 13.32% |
| Gemma-3-12B-It | 14.71% | 27.28% | 11.62% | 28.25% |
| Gemma-3-27B-It | 13.24% | 28.08% | 9.61% | 29.37% |
| Phi-4 | 30.31% | 30.14% | 29.29% | 31.46% |
| Phi-4-Reasoning | 54.94% | 54.68% | 54.39% | 56.66% |
| Phi-4-Reasoning-Plus | 40.69% | 44.02% | 41.38% | 45.10% |
| GPT-4o-mini | 12.40% | 35.23% | 9.84% | 35.96% |
| GPT-4o | 18.21% | 37.84% | 14.99% | 39.00% |
| Claude-3.5-Haiku | 42.46% | 45.10% | 38.64% | 46.71% |
| Claude-3.7-Sonnet | 46.33% | 55.11% | 46.41% | 56.66% |
| Gemini-2.0-Flash | 33.66% | 48.21% | 27.35% | 50.24% |
| Gemini-2.0-Flash-Lite | 24.47% | 29.72% | 20.66% | 30.98% |
| Gemini-2.0-Flash-Thinking | 42.30% | 56.61% | 35.51% | 58.59% |
| Gemini-2.5-Flash | 32.40% | 65.65% | 28.20% | 68.70% |
| Gemini-2.5-Pro | 33.05% | **76.51%** | 30.50% | **80.42%** |

Table 15: Evaluation of code generated by various LLMs on EFFIBENCH-X-JavaScript.

| Model Name ET | MP | MI | Pass@1 | |
|---|---|---|---|---|
| DeepSeek-V3-0324 | 40.18% | 51.22% | 39.75% | 52.17% |
| DeepSeek-R1 | 63.34% | 69.19% | 63.03% | 71.43% |
| Llama-4-Scout-17B-16E-Instruct | 24.70% | 31.20% | 24.61% | 31.46% |
| Llama-4-Maverick-17B-128E-Instruct | 14.10% | 44.75% | 14.45% | 45.10% |
| Qwen3-8B | 52.38% | 54.11% | 52.37% | 55.22% |
| Qwen3-14B | 60.65% | 61.65% | 60.05% | 63.08% |
| Qwen3-32B | **70.29%** | 71.77% | **69.87%** | 73.84% |
| Qwen2.5-Coder-7B-Instruct | 27.25% | 28.78% | 26.88% | 29.05% |
| Qwen2.5-Coder-14B-Instruct | 33.51% | 36.62% | 33.08% | 37.08% |
| Qwen2.5-Coder-32B-Instruct | 36.98% | 40.55% | 36.48% | 41.09% |
| QwQ-32B | 32.79% | 34.61% | 32.42% | 35.47% |
| Gemma-3-4B-It | 10.23% | 18.86% | 10.27% | 19.58% |
| Gemma-3-12B-It | 15.48% | 28.93% | 15.74% | 29.37% |
| Gemma-3-27B-It | 13.60% | 31.67% | 13.80% | 32.26% |
| Phi-4 | 29.37% | 31.44% | 28.90% | 31.78% |
| Phi-4-Reasoning | 52.94% | 53.99% | 52.27% | 55.38% |
| Phi-4-Reasoning-Plus | 42.53% | 44.08% | 42.25% | 44.94% |
| GPT-4o-mini | 12.81% | 34.48% | 13.04% | 34.83% |
| GPT-4o | 24.91% | 49.66% | 26.62% | 50.08% |
| Claude-3.5-Haiku | 38.36% | 48.62% | 38.10% | 49.44% |
| Claude-3.7-Sonnet | 50.15% | 57.33% | 49.48% | 58.59% |
| Gemini-2.0-Flash | 31.97% | 52.82% | 32.10% | 53.61% |
| Gemini-2.0-Flash-Lite | 28.06% | 40.92% | 27.66% | 42.22% |
| Gemini-2.0-Flash-Thinking | 40.30% | 60.92% | 40.29% | 62.44% |
| Gemini-2.5-Flash | 37.09% | 66.03% | 37.25% | 68.22% |
| Gemini-2.5-Pro | 47.26% | **78.22%** | 47.93% | **80.90%** |

Table 16: Evaluation of code generated by various LLMs on EFFIBENCH-X-Ruby.

| Model Name ET | MP | MI | Pass@1 | |
|---|---|---|---|---|
| Open-source LLMs | | | | |
| DeepSeek-V3-0324 | 45.61% | 49.28% | 45.65% | 50.88% |
| DeepSeek-R1 | 64.01% | 66.11% | 63.65% | 69.02% |
| Llama-4-Scout-17B-16E-Instruct | 21.30% | 23.19% | 21.51% | 23.60% |
| Llama-4-Maverick-17B-128E-Instruct | 28.05% | 32.59% | 28.93% | 33.23% |
| Qwen3-8B | 44.44% | 45.99% | 44.35% | 47.35% |
| Qwen3-14B | 52.76% | 53.15% | 52.45% | 54.41% |
| Qwen3-32B | 55.47% | 56.32% | 55.25% | 58.27% |
| Qwen2.5-Coder-7B-Instruct | 21.77% | 22.69% | 21.76% | 22.95% |
| Qwen2.5-Coder-14B-Instruct | 30.32% | 32.53% | 30.43% | 32.91% |
| Qwen2.5-Coder-32B-Instruct | 34.35% | 36.67% | 34.40% | 37.24% |
| QwQ-32B | 28.66% | 29.65% | 28.65% | 30.18% |
| Gemma-3-4B-It | 11.72% | 13.76% | 12.04% | 13.80% |
| Gemma-3-12B-It | 21.37% | 26.26% | 21.68% | 26.81% |
| Gemma-3-27B-It | 25.48% | 31.84% | 25.86% | 32.42% |
| Phi-4 | 26.66% | 28.34% | 26.65% | 28.89% |
| Phi-4-Reasoning | 47.53% | 48.61% | 47.22% | 49.92% |
| Phi-4-Reasoning-Plus | 38.32% | 39.49% | 38.24% | 40.61% |
| GPT-4o-mini | 30.72% | 36.52% | 31.48% | 37.24% |
| GPT-4o | 41.34% | 47.64% | 42.39% | 48.48% |
| Claude-3.5-Haiku | 35.47% | 41.13% | 35.66% | 42.05% |
| Claude-3.7-Sonnet | 49.44% | 52.48% | 49.59% | 53.61% |
| Gemini-2.0-Flash | 35.75% | 44.32% | 35.87% | 45.43% |
| Gemini-2.0-Flash-Lite | 30.32% | 37.20% | 30.27% | 37.88% |
| Gemini-2.0-Flash-Thinking | 43.40% | 52.69% | 43.63% | 53.77% |
| Gemini-2.5-Flash | 57.06% | 63.60% | 57.88% | 65.65% |
| Gemini-2.5-Pro | **65.66%** | **72.16%** | **66.12%** | **74.32%** |

Table 17: Evaluation of code generated by various LLMs on EFFIBENCH-X-Go.

| Model Name ET | MP | MI | Pass@1 | |
|---|---|---|---|---|
| Open-source LLMs | | | | |
| DeepSeek-V3-0324 | 35.49% | 50.75% | 35.71% | 52.97% |
| DeepSeek-R1 | 60.24% | 70.57% | 60.57% | 73.35% |
| Llama-4-Scout-17B-16E-Instruct | 20.84% | 24.89% | 20.89% | 25.52% |
| Llama-4-Maverick-17B-128E-Instruct | 7.12% | 30.20% | 7.12% | 32.10% |
| Qwen3-8B | 42.88% | 46.34% | 43.33% | 47.83% |
| Qwen3-14B | 56.25% | 56.69% | 56.28% | 58.43% |
| Qwen3-32B | **62.13%** | 65.28% | **62.60%** | 67.42% |
| Qwen2.5-Coder-7B-Instruct | 20.59% | 20.72% | 20.60% | 21.35% |
| Qwen2.5-Coder-14B-Instruct | 30.86% | 32.40% | 30.95% | 33.23% |
| Qwen2.5-Coder-32B-Instruct | 37.08% | 39.13% | 37.19% | 40.13% |
| QwQ-32B | 31.13% | 33.25% | 31.41% | 34.35% |
| Gemma-3-4B-It | 6.15% | 15.24% | 6.17% | 16.05% |
| Gemma-3-12B-It | 9.11% | 27.56% | 9.12% | 28.89% |
| Gemma-3-27B-It | 9.06% | 32.41% | 9.08% | 34.35% |
| Phi-4 | 25.45% | 26.68% | 25.50% | 27.29% |
| Phi-4-Reasoning | 41.35% | 42.15% | 41.48% | 42.70% |
| Phi-4-Reasoning-Plus | 31.64% | 32.35% | 31.81% | 33.07% |
| GPT-4o-mini | 8.40% | 32.96% | 8.44% | 34.83% |
| GPT-4o | 12.05% | 35.11% | 12.20% | 37.24% |
| Claude-3.5-Haiku | 32.07% | 43.23% | 32.18% | 44.78% |
| Claude-3.7-Sonnet | 43.48% | 51.52% | 43.67% | 53.77% |
| Gemini-2.0-Flash | 18.18% | 43.94% | 18.19% | 46.07% |
| Gemini-2.0-Flash-Lite | 22.79% | 41.05% | 22.78% | 42.70% |
| Gemini-2.0-Flash-Thinking | 26.21% | 52.72% | 26.23% | 55.22% |
| Gemini-2.5-Flash | 24.08% | 61.09% | 24.17% | 64.21% |
| Gemini-2.5-Pro | 31.50% | **76.11%** | 31.61% | **79.94%** |

Table 18: Evaluation of code generated by various LLMs on the EFFIBENCH-X-functional subset.

| Model Name | ET | MP | MI | Pass@1 |
|---|---|---|---|---|
| Open-source LLMs | | | | |
| DeepSeek-V3-0324 | 45.95% | 58.46% | 45.01% | 59.52% |
| DeepSeek-R1 | **74.53%** | 83.26% | **73.26%** | 85.60% |
| Llama-4-Scout-17B-16E-Instruct | 29.47% | 32.48% | 28.80% | 32.79% |
| Llama-4-Maverick-17B-128E-Instruct | 20.71% | 42.62% | 19.91% | 43.29% |
| Qwen3-8B | 55.88% | 63.69% | 55.83% | 65.09% |
| Qwen3-14B | 72.92% | 74.78% | 72.37% | 76.53% |
| Qwen3-32B | 73.14% | 80.68% | 72.85% | 82.74% |
| Qwen2.5-Coder-7B-Instruct | 28.02% | 28.58% | 27.74% | 28.85% |
| Qwen2.5-Coder-14B-Instruct | 37.78% | 40.15% | 37.15% | 40.53% |
| Qwen2.5-Coder-32B-Instruct | 45.49% | 48.06% | 44.93% | 48.67% |
| QwQ-32B | 34.21% | 37.55% | 34.17% | 38.12% |
| Gemma-3-4B-It | 11.99% | 20.18% | 11.43% | 20.56% |
| Gemma-3-12B-It | 20.37% | 35.25% | 18.80% | 36.19% |
| Gemma-3-27B-It | 21.94% | 41.74% | 20.21% | 42.95% |
| Phi-4 | 30.54% | 32.32% | 29.80% | 32.79% |
| Phi-4-Reasoning | 56.73% | 57.21% | 55.48% | 58.28% |
| Phi-4-Reasoning-Plus | 41.18% | 42.52% | 40.40% | 43.15% |
| Proprietary LLMs | | | | |
| GPT-4o-mini | 18.01% | 37.46% | 17.08% | 38.12% |
| GPT-4o | 24.33% | 44.20% | 23.25% | 45.02% |
| Claude-3.5-Haiku | 44.72% | 54.04% | 43.38% | 55.13% |
| Claude-3.7-Sonnet | 54.91% | 63.57% | 54.42% | 64.89% |
| Gemini-2.0-Flash | 36.46% | 55.27% | 33.90% | 56.56% |
| Gemini-2.0-Flash-Lite | 35.60% | 47.93% | 33.74% | 48.96% |
| Gemini-2.0-Flash-Thinking | 47.60% | 66.95% | 45.23% | 68.39% |
| Gemini-2.5-Flash | 46.42% | 74.12% | 43.79% | 76.48% |
| Gemini-2.5-Pro | 57.41% | **90.04%** | 53.78% | **92.50%** |

Table 19: Evaluation of code generated by various LLMs on the EFFIBENCH-X-standard I/O subset.

| Model Name | ET | MP | MI | Pass@1 |
|---|---|---|---|---|
| DeepSeek-V3-0324 | 33.95% | 43.29% | 32.70% | 45.91% |
| DeepSeek-R1 | 45.68% | 52.99% | 44.42% | 57.60% |
| Llama-4-Scout-17B-16E-Instruct | 15.66% | 22.89% | 15.28% | 23.27% |
| Llama-4-Maverick-17B-128E-Instruct | 11.03% | 29.18% | 10.31% | 30.23% |
| Qwen3-8B | 33.06% | 37.35% | 32.40% | 39.77% |
| Qwen3-14B | 44.12% | 44.19% | 42.22% | 47.60% |
| Qwen3-32B | **49.25%** | 51.33% | **48.00%** | 55.79% |
| Qwen2.5-Coder-7B-Instruct | 20.05% | 21.63% | 19.60% | 22.05% |
| Qwen2.5-Coder-14B-Instruct | 25.28% | 27.12% | 24.22% | 28.19% |
| Qwen2.5-Coder-32B-Instruct | 26.17% | 28.51% | 25.51% | 29.59% |
| QwQ-32B | 28.51% | 33.09% | 28.07% | 35.20% |
| Gemma-3-4B-It | 6.97% | 12.66% | 6.58% | 13.10% |
| Gemma-3-12B-It | 9.79% | 18.61% | 8.88% | 18.83% |
| Gemma-3-27B-It | 10.30% | 21.58% | 9.27% | 22.28% |
| Phi-4 | 25.91% | 26.84% | 24.42% | 28.01% |
| Phi-4-Reasoning | 38.46% | 38.89% | 36.89% | 41.35% |
| Phi-4-Reasoning-Plus | 31.22% | 33.12% | 30.48% | 34.68% |
| GPT-4o-mini | 15.70% | 32.34% | 15.13% | 33.63% |
| GPT-4o | 24.77% | 40.73% | 24.90% | 41.93% |
| Claude-3.5-Haiku | 26.38% | 32.22% | 25.22% | 33.51% |
| Claude-3.7-Sonnet | 39.35% | 43.95% | 38.15% | 45.96% |
| Gemini-2.0-Flash | 23.69% | 37.52% | 22.25% | 39.06% |
| Gemini-2.0-Flash-Lite | 18.68% | 27.55% | 17.43% | 29.12% |
| Gemini-2.0-Flash-Thinking | 28.41% | 42.05% | 26.88% | 44.33% |
| Gemini-2.5-Flash | 33.31% | 54.46% | 31.62% | 58.13% |
| Gemini-2.5-Pro | 36.45% | **58.47%** | 34.76% | **63.92%** |

Table 20: Algorithm Tag Analysis: per-tag efficiency by metric (%).

| Algorithm Tag | MP (%) | ET (%) | MI (%) |
|---|---|---|---|
| string-matching | 90.91 | 86.85 | 86.48 |
| simulation | 89.29 | 80.37 | 77.99 |
| counting | 82.74 | 76.85 | 74.72 |
| hash-table | 76.75 | 71.61 | 70.16 |
| two-pointers | 76.14 | 79.85 | 77.02 |
| enumeration | 72.19 | 71.52 | 70.16 |
| string | 71.56 | 67.85 | 67.08 |
| sliding-window | 69.23 | 59.91 | 58.52 |
| math | 68.53 | 65.65 | 63.72 |
| sorting | 68.52 | 64.86 | 63.54 |
| array | 67.31 | 63.91 | 62.19 |
| graph | 60.44 | 58.96 | 58.20 |
| number-theory | 60.00 | 58.87 | 57.64 |
| binary-search | 58.21 | 52.26 | 51.43 |
| prefix-sum | 54.41 | 50.17 | 48.88 |
| matrix | 53.30 | 52.59 | 51.08 |
| bit-manipulation | 53.13 | 51.98 | 50.57 |
| heap/priority-queue | 46.67 | 44.89 | 43.49 |
| combinatorics | 44.64 | 39.43 | 36.17 |
| greedy | 43.08 | 40.12 | 40.00 |
| dynamic-programming | 23.09 | 21.69 | 20.75 |

functional problems. Claude-3.7-Sonnet performs well (39.35% ET, 45.96% Pass@1), outperforming many other closed-source alternatives.

Across all models, performance on functional problems significantly exceeds that on standard I/O problems, with most models showing 15-30 percentage point higher scores on the functional subset for both efficiency and correctness metrics. This pattern holds regardless of model size or architecture. For instance, DeepSeek-R1 achieves 74.53% ET and 85.60% Pass@1 on functional problems but only 45.68% ET and 57.60% Pass@1 on standard I/O problems. Similarly, Gemini-2.5-Pro shows 57.41% ET and 92.50% Pass@1 versus 36.45% ET and 63.92% Pass@1 on these respective categories.

Despite these absolute differences, the relative performance hierarchy remains consistent across both problem types. DeepSeek-R1, Qwen3-32B, and Gemini-2.5-Pro maintain their leadership positions in their respective metrics across both categories, while smaller models like Gemma-3-4B-It (11.99% ET on functional, 6.97% on standard I/O) consistently rank lower. Both problem types show similar scaling patterns with model size and generation, with larger and newer models generally outperforming their smaller and older counterparts.

The performance gap between functional and standard I/O problems reveals that current LLMs struggle more with optimizing I/O operations, buffer management, and input parsing compared to core algorithmic logic. Standard I/O problems involve more complex implementation aspects, creating additional opportunities for inefficiencies. This finding suggests that enhanced training on I/O patterns and memory management could substantially improve overall code generation capabilities, particularly for solving end-to-end programming problems that require both algorithmic reasoning and efficient implementation details.

## C.8 Algorithm Tag Analysis

We report per-tag efficiency across ET, MP, and MI to reveal capability differences across problem categories. As shown in Table 20, procedural tags such as string-matching, simulation, and counting achieve the highest efficiencies, whereas dynamic-programming and greedy remain challenging across all three metrics.

## C.9 Language Runtime Specifications

Table 21 details the specific Docker images and compilation/execution flags used for each programming language in our sandboxed execution environment. For C++, we utilize gcc:14.2.0-bookworm with optimization level -O2 and address sanitization enabled (-fsanitize=address) to ensure memory safety while maintaining good performance. Java code is executed using openjdk:21-jdk-bookworm without additional flags, leveraging the JVM's default optimization capabilities. For JavaScript, we employ node:22.14.0-bookworm with the –harmony flag to enable the latest ECMAScript features. Ruby evaluations use ruby:3.2.7-bookworm, while Go utilizes golang:1.23.7-bookworm, both with their respective default runtime configurations. Python code runs on python:3.11.11-bookworm, which provides a balanced combination of performance and compatibility with modern Python libraries. All these environments are consistent across evaluations, ensuring fair comparisons between human reference solutions and LLM-generated code while capturing real-world performance characteristics for each language.

Table 21: Language Runtime Specifications and Flags

| Language | Docker Image | Compilation/Execution Flags |
| --- | --- | --- |
| C++ | gcc:14.2.0-bookworm | -O2, -fsanitize=address |
| Java | openjdk:21-jdk-bookworm | - |
| JavaScript | node:22.14.0-bookworm | –harmony |
| Ruby | ruby:3.2.7-bookworm | - |
| Go | golang:1.23.7-bookworm | - |
| Python3 | python:3.11.11-bookworm | - |

## C.10 Stochastic Sampling and Multi-Solution Efficiency (`pass@5`)

To examine the impact of stochastic decoding on code efficiency, we extended our evaluation beyond greedy decoding (`pass@1`) to a multi-sampling setup (`pass@5`). For each representative model, we generated five candidate solutions per task at `temperature = 0.8` and evaluated them under two perspectives: (1) the most efficient correct solution, and (2) the average efficiency across all correct solutions.

**Most Efficient Solutions** (`pass@5`). For each task with at least one correct solution among the five samples, we selected the single most efficient correct solution for scoring. As shown in Table 23, this provides an upper bound on the efficiency a model can achieve with multiple attempts. DeepSeek-V2 achieves the highest integral score (43.04%), followed by Claude 3.5 Sonnet (32.97%). These results indicate that allowing multiple attempts improves both pass rate and overall efficiency.

**Average Efficiency of Correct Solutions** (`pass@5`). To provide a more conservative estimate, we also computed the average efficiency across all functionally correct solutions within the five samples (Table 24). While the absolute scores are slightly lower, the relative ranking among models remains consistent. Stochastic sampling thus improves both correctness and efficiency without altering comparative trends across models.

## C.11 Unexpected Regression in Reasoning-Enhanced Models:

An intriguing observation in our results is the performance regression seen in Phi-4-Reasoning-Plus compared to Phi-4-Reasoning. While Phi-4-Reasoning achieves respectable performance metrics

Table 22: Rate of LLM-generated responses that do not extract code, presented per model.

| Model Name | all | C++ | Java | JavaScript | Ruby | Go | Python |
| --- | --- | --- | --- | --- | --- | --- | --- |
| DeepSeek-V3-0324 | 32/3738 (0.86%) | 5/623 (0.80%) | 4/623 (0.64%) | 5/623 (0.80%) | 8/623 (1.28%) | 4/623 (0.64%) | 6/623 (0.96%) |
| DeepSeek-R1 | 6/3738 (0.16%) | 2/623 (0.32%) | 1/623 (0.16%) | 1/623 (0.16%) | 1/623 (0.16%) | 1/623 (0.16%) | 0/623 (0.00%) |
| Gemini-2.0-Flash | 53/3738 (1.42%) | 2/623 (0.32%) | 2/623 (0.32%) | 5/623 (0.80%) | 10/623 (1.61%) | 13/623 (2.09%) | 21/623 (3.37%) |
| Gemini-2.0-Flash-Thinking | 32/3738 (0.86%) | 2/623 (0.32%) | 3/623 (0.48%) | 7/623 (1.12%) | 7/623 (1.12%) | 8/623 (1.28%) | 5/623 (0.80%) |
| Gemini-2.5-Flash | 78/3738 (2.09%) | 12/623 (1.93%) | 17/623 (2.73%) | 13/623 (2.09%) | 11/623 (1.77%) | 15/623 (2.41%) | 10/623 (1.61%) |
| Gemini-2.5-Pro | 2/3738 (0.05%) | 0/623 (0.00%) | 0/623 (0.00%) | 0/623 (0.00%) | 0/623 (0.00%) | 2/623 (0.32%) | 0/623 (0.00%) |
| Phi-4-reasoning | 1151/3738 (30.79%) | 254/623 (40.77%) | 161/623 (25.84%) | 176/623 (28.25%) | 183/623 (29.37%) | 180/623 (28.89%) | 197/623 (31.62%) |
| Phi-4-reasoning-plus | 1957/3738 (52.35%) | 371/623 (59.55%) | 297/623 (47.67%) | 299/623 (47.99%) | 315/623 (50.56%) | 327/623 (52.49%) | 348/623 (55.86%) |

Table 23: Most efficient solution scores under stochastic decoding (pass@5).

| Model | Runtime Score (%) | Memory Score (%) | Integral Score (%) | Pass Rate (%) |
|---|---|---|---|---|
| DeepSeek-V2 | 41.80 | 44.49 | 43.04 | 46.07 |
| Qwen-32B | 27.64 | 29.13 | 28.61 | 29.86 |
| GPT-4o | 26.15 | 28.09 | 26.89 | 28.57 |
| Claude 3.5 Sonnet | 31.88 | 34.83 | 32.97 | 35.47 |

Table 24: Average efficiency scores of correct solutions under stochastic decoding (pass@5).

| Model | Average Runtime Score (%) | Average Memory Score (%) | Average Integral Score (%) |
|---|---|---|---|
| DeepSeek-V2 | 34.88 | 35.76 | 34.28 |
| Qwen-32B | 22.03 | 22.80 | 21.93 |
| GPT-4o | 46.89 | 47.74 | 46.88 |
| Claude 3.5 Sonnet | 28.28 | 29.09 | 28.16 |

across languages (ET 48.37%, Pass@1 50.54%), its enhanced counterpart Phi-4-Reasoning-Plus shows notably lower performance (ET 37.55%, Pass@1 38.65%). As shown in Table 22, this regression can be largely attributed to the inability of Phi-4-Reasoning-Plus to properly generate extractable code in many cases, with a concerning 52.35% of responses lacking proper code extraction across all languages, compared to just 30.79% for Phi-4-Reasoning. This problem is particularly pronounced in C++ (59.55% non-extractable responses) and Python (55.86%). In contrast, high-performing models like DeepSeek-R1 and Gemini-2.5-Pro have minimal code extraction issues (0.16% and 0.05%, respectively), demonstrating their superior ability to produce well-structured outputs. These findings suggest that for intermediate model sizes like Phi-4 (16B), adding more complex reasoning capabilities might actually interfere with the model's ability to focus on the core task of generating functional code. The model appears to over-index on explanation and verbosity at the expense of concise, executable solutions. This stands in contrast to reasoning enhancements in larger models, where reasoning capabilities complement rather than detract from code generation performance. Our findings indicate that effective reasoning for code generation may have a model size threshold below which the added complexity becomes detrimental rather than beneficial, aligning with findings on Chain-of-Thought reasoning documented by Wei et al. [68], which confirm that model size significantly impacts the reasoning abilities of LLMs.

### C.12 Profile with Tighter Time Limits

To assess the robustness of our execution time configuration, we re-evaluated all models under stricter time limits by halving the default cutoff from 10 seconds to 5 seconds. As shown in Table 25, the performance of Claude-3.7-Sonnet remains nearly identical across different cutoff thresholds (10s, 7s, and 5s), with only a negligible drop observed at a very aggressive 3s limit. Table 26 further demonstrates that the overall model ranking remains entirely unchanged between the 10s and 5s configurations. These results confirm that our original 10-second limit is a robust and fair choice, ensuring that slower models are not unduly penalized while accurately reflecting realistic runtime behavior.

Table 25: Performance of Claude 3.7 Sonnet at various execution time limits.

| Time Limit | ET (%) |
|---|---|
| 10s (Original) | 53.63 |
| 7s | 53.63 |
| 5s | 53.63 |
| 3s | 53.50 |

Table 26: Ranking of models under different execution time cutoffs.

| Model | ET @ 10s Cutoff (%) | ET @ 5s Cutoff (%) | Change (%) |
|---|---|---|---|
| Gemini-2.5-Pro | 64.92 | 64.92 | 0.00 |
| Gemini-2.5-Flash | 58.95 | 58.88 | -0.07 |
| Claude-3.7-Sonnet | 53.63 | 53.63 | 0.00 |
| GPT-4o | 40.12 | 40.12 | 0.00 |
| Claude-3.5-Haiku | 35.97 | 35.95 | -0.02 |
| GPT-4o-mini | 28.63 | 28.62 | -0.01 |

Table 27: Evaluation of code generated by various LLMs on EFFIBENCH-X-Aizu.

| Model Name | Execution Time (ET) (%) | Memory Peak (MP) (%) | Memory Integral (MI) (%) | Pass@1 (%) |
|---|---|---|---|---|
| DeepSeek-V3-0324 | 25.81% | 31.38% | 24.61% | 34.34% |
| DeepSeek-R1 | 36.09% | 42.02% | 34.96% | 45.96% |
| Llama-4-Scout-17B-16E-Instruct | 6.83% | 10.73% | 7.01% | 11.11% |
| Llama-4-Maverick-17B-128E-Instruct | 5.05% | 12.68% | 4.66% | 14.14% |
| Qwen3-8B | 18.69% | 21.02% | 18.07% | 22.22% |
| Qwen3-14B | 24.87% | 24.70% | 23.30% | 27.27% |
| Qwen3-32B | 33.96% | 34.82% | 32.62% | 38.89% |
| Qwen2.5-Coder-7B-Instruct | 6.18% | 6.95% | 6.03% | 7.07% |
| Qwen2.5-Coder-14B-Instruct | 10.82% | 11.20% | 10.41% | 12.12% |
| Qwen2.5-Coder-32B-Instruct | 9.78% | 10.37% | 9.15% | 11.62% |
| QwQ-32B | 27.44% | 30.92% | 26.53% | 33.33% |
| Gemma-3-4B-It | 2.70% | 4.37% | 2.61% | 4.55% |
| Gemma-3-12B-It | 2.57% | 4.50% | 2.58% | 4.55% |
| Gemma-3-27B-It | 4.52% | 8.32% | 4.52% | 8.59% |
| Phi-4 | 12.03% | 12.28% | 10.95% | 13.64% |
| Phi-4-Reasoning | 23.78% | 24.26% | 23.27% | 25.76% |
| Phi-4-Reasoning-Plus | 14.39% | 14.86% | 13.90% | 15.66% |
| GPT-4o-mini | 8.96% | 18.02% | 8.90% | 18.69% |
| GPT-4o | 14.35% | 25.06% | 14.75% | 25.76% |
| Claude-3.5-Haiku | 14.64% | 17.71% | 14.06% | 18.69% |
| Claude-3.7-Sonnet | 26.14% | 28.83% | 25.53% | 30.81% |
| Gemini-2.0-Flash | 13.87% | 19.73% | 12.75% | 21.72% |
| Gemini-2.0-Flash-Lite | 10.74% | 15.65% | 10.62% | 16.67% |
| Gemini-2.0-Flash-Thinking | 32.08% | 46.42% | 30.86% | 50.00% |
| Gemini-2.5-Flash | 20.74% | 34.54% | 19.83% | 36.36% |
| Gemini-2.5-Pro | **37.03%** | **58.70%** | **36.94%** | **65.15%** |

## C.13 Website-Level Results

We provide the website-level code efficiency results in Tables 27 to 31.

## C.14 Data Contamination Analysis

Following recent practice in benchmark construction, we assessed potential data contamination in EFFIBENCH-X using two complementary families of methods: (i) strict $n$-gram overlap checks with $n \in \{20, 30\}$ between our 623 problems and public corpora; and (ii) an embedding-based nearest-neighbor search that flags pairs with cosine similarity $> 0.90$. Table 32 summarizes the findings. Across all checks, we observe minimal contamination: the most stringent embedding search identifies only 3/623 tasks (0.48%), while $n$-gram overlap yields 11/623 (1.77%) at $n = 20$ and 6/623 (0.96%) at $n = 30$. These results confirm that EFFIBENCH-X provides a clean and reliable basis for evaluating code-efficiency generation.

## C.15 Hardware Variation

To validate the robustness and generalizability of our benchmark, we conducted experiments across different hardware platforms. As shown in Table 33, the relative efficiency ranking of models (*Gemini-2.5-Pro* > *DeepSeek-R1* > *GPT-4o*) remained consistent on both AMD and Intel systems. Furthermore, our normalization method effectively abstracts hardware-level differences, as the final integral efficiency scores for each model varied by less than 3% between platforms. These findings confirm that our methodology provides a stable, hardware-agnostic measure of model efficiency.

Table 28: Evaluation of code generated by various LLMs on EFFIBENCH-X-AtCoder.

| Model Name | Execution Time (ET) (%) | Memory Peak (MP) (%) | Memory Integral (MI) (%) | Pass@1 (%) |
|---|---|---|---|---|
| DeepSeek-V3-0324 | 29.97% | 37.61% | 28.06% | 41.16% |
| DeepSeek-R1 | 37.36% | 42.25% | 35.33% | 48.21% |
| Llama-4-Scout-17B-16E-Instruct | 10.78% | 19.41% | 10.41% | 19.80% |
| Llama-4-Maverick-17B-128E-Instruct | 9.69% | 26.13% | 9.00% | 27.29% |
| Qwen3-8B | 25.09% | 28.46% | 23.77% | 31.32% |
| Qwen3-14B | 35.39% | 35.32% | 32.87% | 39.60% |
| Qwen3-32B | **42.85%** | 42.77% | **40.72%** | 48.77% |
| Qwen2.5-Coder-7B-Instruct | 17.18% | 19.00% | 16.55% | 19.57% |
| Qwen2.5-Coder-14B-Instruct | 23.85% | 25.90% | 22.33% | 27.40% |
| Qwen2.5-Coder-32B-Instruct | 24.64% | 27.08% | 23.63% | 28.52% |
| QwQ-32B | 30.65% | 35.81% | 29.86% | 39.15% |
| Gemma-3-4B-It | 6.21% | 11.23% | 5.87% | 11.52% |
| Gemma-3-12B-It | 8.74% | 16.67% | 7.85% | 16.89% |
| Gemma-3-27B-It | 9.05% | 19.35% | 7.92% | 20.13% |
| Phi-4 | 21.97% | 22.93% | 20.44% | 24.50% |
| Phi-4-Reasoning | 31.30% | 31.15% | 28.98% | 34.45% |
| Phi-4-Reasoning-Plus | 22.55% | 23.81% | 21.69% | 25.73% |
| GPT-4o-mini | 14.11% | 28.71% | 13.41% | 30.54% |
| GPT-4o | 24.02% | 38.21% | 23.75% | 39.93% |
| Claude-3.5-Haiku | 23.87% | 29.35% | 22.31% | 31.21% |
| Claude-3.7-Sonnet | 35.20% | 39.39% | 33.24% | 42.17% |
| Gemini-2.0-Flash | 21.73% | 34.41% | 20.17% | 36.47% |
| Gemini-2.0-Flash-Lite | 15.94% | 23.47% | 14.63% | 25.28% |
| Gemini-2.0-Flash-Thinking | 24.17% | 35.83% | 22.43% | 38.81% |
| Gemini-2.5-Flash | 27.56% | 44.72% | 25.57% | 49.78% |
| Gemini-2.5-Pro | 35.24% | **55.78%** | 32.17% | **63.42%** |

Table 29: Evaluation of code generated by various LLMs on EFFIBENCH-X-CodeChef.

| Model Name | Execution Time (ET) (%) | Memory Peak (MP) (%) | Memory Integral (MI) (%) | Pass@1 (%) |
|---|---|---|---|---|
| DeepSeek-V3-0324 | 42.74% | 55.97% | 42.48% | 57.17% |
| DeepSeek-R1 | 62.31% | 73.85% | 62.22% | 76.88% |
| Llama-4-Scout-17B-16E-Instruct | 26.57% | 33.12% | 25.91% | 33.51% |
| Llama-4-Maverick-17B-128E-Instruct | 15.46% | 40.29% | 14.56% | 41.04% |
| Qwen3-8B | 50.15% | 56.55% | 50.55% | 58.78% |
| Qwen3-14B | 64.57% | 65.03% | 63.54% | 67.56% |
| Qwen3-32B | **64.76%** | 70.47% | **64.92%** | 72.94% |
| Qwen2.5-Coder-7B-Instruct | 30.68% | 32.29% | 30.39% | 32.62% |
| Qwen2.5-Coder-14B-Instruct | 33.08% | 35.16% | 32.63% | 35.66% |
| Qwen2.5-Coder-32B-Instruct | 35.24% | 38.19% | 35.08% | 38.71% |
| QwQ-32B | 26.63% | 30.93% | 26.89% | 31.18% |
| Gemma-3-4B-It | 10.01% | 18.39% | 9.43% | 19.18% |
| Gemma-3-12B-It | 14.46% | 27.28% | 13.19% | 27.60% |
| Gemma-3-27B-It | 14.39% | 30.07% | 13.09% | 30.82% |
| Phi-4 | 38.42% | 39.60% | 36.88% | 40.14% |
| Phi-4-Reasoning | 54.20% | 55.46% | 53.32% | 56.99% |
| Phi-4-Reasoning-Plus | 50.18% | 53.55% | 49.49% | 54.84% |
| GPT-4o-mini | 21.15% | 43.90% | 20.54% | 44.62% |
| GPT-4o | 30.17% | 51.51% | 30.67% | 52.15% |
| Claude-3.5-Haiku | 36.49% | 44.34% | 35.70% | 44.98% |
| Claude-3.7-Sonnet | 51.29% | 57.10% | 50.99% | 58.06% |
| Gemini-2.0-Flash | 30.72% | 49.32% | 29.30% | 50.00% |
| Gemini-2.0-Flash-Lite | 25.37% | 37.61% | 23.88% | 39.07% |
| Gemini-2.0-Flash-Thinking | 34.40% | 51.09% | 33.00% | 51.97% |
| Gemini-2.5-Flash | 46.88% | **76.87%** | 45.26% | **79.21%** |
| Gemini-2.5-Pro | 38.35% | 62.43% | 38.05% | 64.52% |

Table 30: Evaluation of code generated by various LLMs on EFFIBENCH-X-Codeforces.

| Model Name | Execution Time (ET) (%) | Memory Peak (MP) (%) | Memory Integral (MI) (%) | Pass@1 (%) |
|---|---|---|---|---|
| DeepSeek-V3-0324 | 38.47% | 49.32% | 37.71% | 50.00% |
| DeepSeek-R1 | 46.62% | 55.19% | 45.52% | 56.67% |
| Llama-4-Scout-17B-16E-Instruct | 16.13% | 19.72% | 16.23% | 20.00% |
| Llama-4-Maverick-17B-128E-Instruct | 9.50% | 25.88% | 9.02% | 26.67% |
| Qwen3-8B | 40.19% | 45.15% | 39.54% | 46.67% |
| Qwen3-14B | 47.63% | 46.83% | 45.74% | 48.33% |
| Qwen3-32B | **50.69%** | 55.40% | **49.73%** | 56.67% |
| Qwen2.5-Coder-7B-Instruct | 9.72% | 10.00% | 9.32% | 10.00% |
| Qwen2.5-Coder-14B-Instruct | 21.80% | 23.12% | 19.72% | 23.33% |
| Qwen2.5-Coder-32B-Instruct | 18.88% | 19.67% | 18.56% | 20.00% |
| QwQ-32B | 17.70% | 19.94% | 17.53% | 20.00% |
| Gemma-3-4B-It | 4.13% | 8.08% | 3.92% | 8.33% |
| Gemma-3-12B-It | 5.93% | 13.33% | 4.90% | 13.33% |
| Gemma-3-27B-It | 10.03% | 19.67% | 9.63% | 20.00% |
| Phi-4 | 14.12% | 14.55% | 12.28% | 15.00% |
| Phi-4-Reasoning | 47.28% | 48.47% | 46.81% | 50.00% |
| Phi-4-Reasoning-Plus | 39.58% | 42.22% | 39.44% | 43.33% |
| GPT-4o-mini | 11.06% | 26.26% | 11.14% | 26.67% |
| GPT-4o | 20.15% | 29.67% | 21.94% | 30.00% |
| Claude-3.5-Haiku | 8.52% | 10.00% | 8.04% | 10.00% |
| Claude-3.7-Sonnet | 33.83% | 39.57% | 33.63% | 40.00% |
| Gemini-2.0-Flash | 19.79% | 33.00% | 19.10% | 33.33% |
| Gemini-2.0-Flash-Lite | 23.41% | 34.11% | 21.65% | 35.00% |
| Gemini-2.0-Flash-Thinking | 23.84% | 36.28% | 23.01% | 36.67% |
| Gemini-2.5-Flash | 34.35% | 57.09% | 33.78% | 58.33% |
| Gemini-2.5-Pro | 34.91% | **60.99%** | 35.66% | **61.67%** |

Table 31: Evaluation of code generated by various LLMs on EFFIBENCH-X-LeetCode.

| Model Name | Execution Time (ET) (%) | Memory Peak (MP) (%) | Memory Integral (MI) (%) | Pass@1 (%) |
|---|---|---|---|---|
| Open-source LLMs | | | | |
| DeepSeek-V3-0324 | 45.95% | 58.46% | 45.01% | 59.52% |
| DeepSeek-R1 | **74.53%** | 83.26% | **73.26%** | 85.60% |
| Llama-4-Scout-17B-16E-Instruct | 29.47% | 32.48% | 28.80% | 32.79% |
| Llama-4-Maverick-17B-128E-Instruct | 20.71% | 42.62% | 19.91% | 43.29% |
| Qwen3-8B | 55.88% | 63.69% | 55.83% | 65.09% |
| Qwen3-14B | 72.92% | 74.78% | 72.37% | 76.53% |
| Qwen3-32B | 73.14% | 80.68% | 72.85% | 82.74% |
| Qwen2.5-Coder-7B-Instruct | 28.02% | 28.58% | 27.74% | 28.85% |
| Qwen2.5-Coder-14B-Instruct | 37.78% | 40.15% | 37.15% | 40.53% |
| Qwen2.5-Coder-32B-Instruct | 45.49% | 48.06% | 44.93% | 48.67% |
| QwQ-32B | 34.21% | 37.55% | 34.17% | 38.12% |
| Gemma-3-4B-It | 11.99% | 20.18% | 11.43% | 20.56% |
| Gemma-3-12B-It | 20.37% | 35.25% | 18.80% | 36.19% |
| Gemma-3-27B-It | 21.94% | 41.74% | 20.21% | 42.95% |
| Phi-4 | 30.54% | 32.32% | 29.80% | 32.79% |
| Phi-4-Reasoning | 56.73% | 57.21% | 55.48% | 58.28% |
| Phi-4-Reasoning-Plus | 41.18% | 42.52% | 40.40% | 43.15% |
| GPT-4o-mini | 18.01% | 37.46% | 17.08% | 38.12% |
| GPT-4o | 24.33% | 44.20% | 23.25% | 45.02% |
| Claude-3.5-Haiku | 44.72% | 54.04% | 43.38% | 55.13% |
| Claude-3.7-Sonnet | 54.91% | 63.57% | 54.42% | 64.89% |
| Gemini-2.0-Flash | 36.46% | 55.27% | 33.90% | 56.56% |
| Gemini-2.0-Flash-Lite | 35.60% | 47.93% | 33.74% | 48.96% |
| Gemini-2.0-Flash-Thinking | 47.60% | 66.95% | 45.23% | 68.39% |
| Gemini-2.5-Flash | 46.42% | 74.12% | 43.79% | 76.48% |
| Gemini-2.5-Pro | 57.41% | **90.04%** | 53.78% | **92.50%** |

Table 32: Contamination analysis on EFFIBENCH-X (623 tasks).

| Method | Parameters | Contaminated Tasks | Rate (%) |
|---|---|---|---|
| N-gram Overlap | $n = 20$ | 11 / 623 | 1.77 |
| N-gram Overlap | $n = 30$ | 6 / 623 | 0.96 |
| Embedding Search | Cosine similarity $> 0.90$ | 3 / 623 | 0.48 |

Table 33: Efficiency comparison across different hardware platforms.

| Hardware | Model | Runtime | Memory | Integral |
|----------|-------|---------|--------|----------|
| **AMD EPYC** | Gemini-2.5-Pro | 56.83% | 57.45% | 54.13% |
| | DeepSeek-R1 | 48.92% | 49.44% | 47.03% |
| | GPT-4o | 39.04% | 39.00% | 38.59% |
| **Intel Xeon** | Gemini-2.5-Pro | 57.00% | 58.54% | 54.86% |
| | DeepSeek-R1 | 52.08% | 53.06% | 49.93% |
| | GPT-4o | 39.46% | 40.80% | 38.93% |

Table 34: Performance of single-shot generation under different prompt strategies.

| Prompt Strategy | Runtime Pass Rate (%) | Memory Pass Rate (%) | Integral Score (%) |
|-----------------|----------------------|----------------------|--------------------|
| Baseline Prompt | 27.12 | 36.21 | 28.26 |
| aggressive_opt | 31.36 | 31.34 | 31.24 |
| ecco | 31.99 | 31.49 | 31.73 |
| hardware_hint | 30.64 | 30.52 | 30.63 |
| eco_aware | 32.40 | 32.20 | 32.42 |

## C.16 Prompt Design and Iterative Refinement

To investigate how prompting strategies affect single-shot code generation and iterative improvement, we conducted two complementary experiments extending our main study. The first experiment evaluated the impact of different prompt designs on single-shot efficiency generation, while the second assessed the benefits of iterative feedback-based refinement.

**Single-Shot Generation with Advanced Prompts.** We examined several advanced prompts designed to enrich contextual grounding, including persona-style optimization (*aggressive_opt*), hardware-aware instructions (*hardware_hint*), and explicit efficiency goals (*ecco*, *eco_aware*). As shown in Table 34, all advanced prompts outperform the baseline configuration. In particular, the *eco_aware* prompt—framing efficiency as minimizing computational and energy cost—achieves the best overall performance, with runtime and memory pass rates of 32.40% and 32.20%, respectively. These results demonstrate that providing efficiency-oriented context enhances the model's ability to generate optimized code.

**Iterative Refinement with Performance Feedback.** Our second experiment introduces iterative refinement through feedback, where the model adjusts its output based on detailed performance diagnostics from previous generations. We adopt EFFILEARNER as the framework, providing line-level runtime and memory profiles to guide optimization. As summarized in Table 35, fine-grained feedback proves critical: while unsupervised self-refinement yields minimal improvement, comprehensive feedback from EFFILEARNER achieves the best results, with a runtime pass rate of 39.19% and memory pass rate of 40.59%. This confirms that LLMs benefit most from explicit, interpretable feedback that identifies not only the presence but also the source of inefficiency.

Table 35: Performance of iterative refinement under different feedback strategies.

| Refinement Strategy | Runtime Pass Rate (%) | Memory Pass Rate (%) | Integral Score (%) |
|---------------------|----------------------|----------------------|--------------------|
| Unsupervised Self-Refine | 34.39 | 36.63 | 34.17 |
| Result Aware | 37.16 | 37.93 | 37.24 |
| Line Profiler | 35.83 | 36.64 | 35.02 |
| Memory Profiler | 35.89 | 40.59 | 38.33 |
| EffiLearner (Full Feedback) | 39.19 | 40.59 | 38.83 |

### C.17 Additional Related Work

The increasing popularity of LLMs for code generation has coincided with the growing availability of open-source code repositories and the need to boost developer productivity. Initial efforts focused on training models specifically for coding tasks, such as CodeT5 [67], AlphaCode [46], CodeGen [53], InCoder [25], StarCoder [45], SantaCoder [5], QwenCoder [37], and DeepSeek-Coder [20]. Contrastingly, models such as Codex [16], CodeLlama [59], Magicoder [70], and WizardCoder [50] represent a subsequent stride, being fine-tuned from foundation models [13, 64]. These code LLMs have been applied to various tasks, including code generation [16, 19, 11, 29], program repair [27, 41], automated testing [44, 21], code translation [60, 2], type prediction [52, 69], and code summarization [28, 3]. While LLMs have achieved impressive results in code generation tasks like HumanEval [16] and MBPP [11], their efficiency has received less attention. Recent studies [61, 32, 54] have shown that LLM-generated code exhibits lower efficiency in terms of execution time and memory usage compared to canonical solutions. These findings highlight the need for further research and development to improve the efficiency of LLM-generated code. In this work, we propose the first large-scale multi-language benchmark specifically designed for robust efficiency evaluation of LLM-generated code across different programming languages.

## D    Case Studies

**Efficient Example 1:** As shown in Figure 3, we provide a Python case example where the solution generated by `DeepSeek-R1` closely matches the human-written canonical solution, both in structure and runtime efficiency (ratio = 0.985). The problem requires computing the sum of absolute differences between adjacent characters' ASCII values in a given string. The `DeepSeek-R1` implementation uses a forward loop from index 1, calculating `abs(ord(s[i]) - ord(s[i-1]))`, while the canonical solution performs the same calculation from index 0 using `abs(ord(s[i]) - ord(s[i+1]))`. These two approaches are functionally equivalent due to the symmetric nature of the absolute difference, and both operate in linear time complexity $\mathcal{O}(n)$ with constant space usage. This convergence highlights the LLM's ability to infer optimal solutions not only from problem descriptions but also from general coding patterns. Unlike in worst-case examples where LLMs diverge in algorithm design or introduce unnecessary overhead, this case demonstrates a high-quality, generalized solution that mirrors expert-level efficiency and readability.

**Efficient Example 2:** As shown in Figure 4, we provide a C++ best-case example in which the code generated by Claude-3.7-Sonnet almost perfectly replicates the canonical solution. The task is to return the XOR of all numbers that appear exactly twice in an array. Both implementations adopt the same high-level strategy: (1) Count the frequency of each element in the input array; (2) Iterate through the frequency map and apply the XOR operation only to elements with a frequency of two. The canonical solution employs a `map`, which maintains ordering but results in slightly higher overhead due to its `O(log N)` access and insertion times. In contrast, the LLM-generated code optimizes performance by utilizing an `unordered_map`, which allows for average-case `O(1)` access and insertion, demonstrating not only correctness but also superior performance awareness. This case exemplifies a high-efficiency general solution, where the LLM demonstrates an understanding of optimal data structures and adheres to clean, human-like code organization. It highlights Claude's capability to generate competitive solutions that are not only correct but also efficient, often rivaling or even surpassing human-written implementations in specific scenarios.

**Worst Case Example 1:** As shown in Figure 5 and Figure 6, the solution generated by Claude-3.7-Sonnet for counting the number of good strings—which can be rearranged to contain ``leet'' as a substring—exemplifies a clear inefficiency in both memory and runtime performance. The task requires counting strings of length $n$ that contain at least one character `l`, at least two `e`'s, and one `t`.

Claude-3.7-Sonnet approaches this problem using a 4-dimensional dynamic programming (DP) table. Specifically, the state `dp[i][l][e][t]` records the number of strings of length $i$ with $l$ occurrences of `l`, $e$ of `e`, and $t$ of `t`, capped respectively at 1, 2, and 1. This results in a state space of size $O(n \times 2 \times 3 \times 2)$, and the model updates each state at every iteration. While the solution is logically correct, it suffers from redundant state expansions and excessive memory usage, especially for large $n$. The model does not employ pruning strategies, prefix-counting optimizations, or algebraic transformations, and instead falls back on a simulation-based enumeration of state transitions.

### DeepSeek-R1

```python
class Solution:
    def scoreOfString(self, s: str) ->
    ↪ int:
        score = 0
        for i in range(1, len(s)):
            score += abs(ord(s[i]) -
            ↪ ord(s[i-1]))
        return score
```

### Human Expert

```python
class Solution:
    def scoreOfString(self, s: str) ->
    ↪ int:
        ans = 0
        n = len(s)
        for i in range(n - 1):
            ans += abs(ord(s[i]) -
            ↪ ord(s[i+1]))
        return ans
```

Figure 3: This figure shows a Python problem where DeepSeek-R1 generated code with efficiency nearly matching the human expert solution (ratio = 0.985). Both calculate adjacent character ASCII differences using equivalent linear-time algorithms with constant space complexity, demonstrating that well-trained LLMs can identify optimal approaches matching expert solutions.

**Claude-3.7-Sonnet**

```cpp
class Solution {
public:
    int duplicateNumbersXOR(vector<int>&
    ↪ nums) {
        unordered_map<int, int> frequency;
        for (int num : nums) {
            frequency[num]++;
        }
        int result = 0;
        for (const auto& pair : frequency)
        ↪ {
            if (pair.second == 2) {
                result ^= pair.first;
            }
        }
        return result;
    }
};
```

**Human Expert**

```cpp
class Solution {
public:
    int duplicateNumbersXOR(vector<int>&
    ↪ nums) {
        map<int,int> mp;
        for(int i : nums) {
            mp[i]++;
        }

        int ans = 0;
        for(auto i : mp) {
            if(i.second == 2) {
                ans ^= i.first;
            }
        }
        return ans;
    }
};
```

Figure 4: This figure shows a C++ problem where Claude-3.7-Sonnet's solution achieves exceptional efficiency by choosing an optimal data structure.

In contrast, the human expert adopts a concise and efficient analytical approach using the *Inclusion-Exclusion Principle*. Rather than enumerating each valid string, it directly subtracts the number of invalid strings (e.g., those missing required characters) from the total $26^n$. By leveraging combinatorial identities and fast modular exponentiation, it computes the final result in $O(\log n)$ time with constant space.

This example clearly illustrates a broader pattern observed in LLM-generated code: while often logically valid, such solutions tend to lack mathematical abstraction and optimization, opting instead for generic enumeration techniques. In this case, the Claude-3.7-Sonnet solution was approximately **33× slower** and significantly more memory-intensive compared to the canonical one, underscoring its limitations in recognizing mathematically structured problem-solving strategies.

## Claude-3.7-Sonnet

```python
class Solution:
    def stringCount(self, n: int) -> int:
        MOD = 10**9 + 7
        dp = [[[[0 for _ in range(2)] for _ in range(3)] for _ in range(2)] for _ in
        ↪  range(n+1)]
        dp[0][0][0][0] = 1
        for i in range(n):
            for l in range(2):
                for e in range(3):
                    for t in range(2):
                        count = dp[i][l][e][t]
                        if count == 0:
                            continue
                        dp[i+1][min(l+1, 1)][e][t] = (dp[i+1][min(l+1, 1)][e][t] + count) %
                        ↪  MOD
                        dp[i+1][l][min(e+1, 2)][t] = (dp[i+1][l][min(e+1, 2)][t] + count) %
                        ↪  MOD
                        dp[i+1][l][e][min(t+1, 1)] = (dp[i+1][l][e][min(t+1, 1)] + count) %
                        ↪  MOD
                        dp[i+1][l][e][t] = (dp[i+1][l][e][t] + count * 23) % MOD
        total = pow(26, n, MOD)
        bad = 0
        for l in range(2):
            for e in range(3):
                for t in range(2):
                    if l == 1 and e == 2 and t == 1:
                        continue
                    bad = (bad + dp[n][l][e][t]) % MOD
        return (total - bad) % MOD
```

Figure 5: This figure shows Claude-3.7-Sonnet's inefficient solution using 4D dynamic programming to count strings containing "leet."

```python
class Solution:
    def stringCount(self, n: int) -> int:
        if n < 4:
            return 0   # Cannot fit 'leet' with fewer than 4 characters

        M = 10**9 + 7
        total = pow(26, n, M)   # All possible lowercase strings

        # Inclusion-Exclusion: Subtract cases that fail to meet 'leet' criteria

        # No 'l'
        sub = pow(25, n, M)
        # No 't'
        sub = (sub + pow(25, n, M)) % M
        # 0 or 1 'e'
        sub = (sub + pow(25, n, M) + n * pow(25, n - 1, M)) % M

        # No 'l' and no 't'
        sub = (sub - pow(24, n, M)) % M
        # No 'l' and <=1 'e'
        sub = (sub - pow(24, n, M) - n * pow(24, n - 1, M)) % M
        sub = (sub - pow(24, n, M) - n * pow(24, n - 1, M)) % M
        sub = (sub + pow(23, n, M) + n * pow(23, n - 1, M)) % M
        return (total - sub) % M
```

Figure 6: This figure presents the human expert's elegant solution using the Inclusion-Exclusion Principle, which directly calculates the answer by subtracting invalid configurations from the total possible strings.

