# OpenReview forum: "EffiBench-X: A Multi-Language Benchmark for Measuring Efficiency of LLM-Generated Code"
_NeurIPS.cc/2025/Datasets_and_Benchmarks_Track — NeurIPS 2025 Datasets and Benchmarks Track poster_

### Official Review · Reviewer_2u2z · 2025-07-01

**Rating:** 4
**Confidence:** 3

**Summary:**

The paper presents EffiBench-X, a new benchmark designed to evaluate the efficiency (rather than just functional correctness) of code generated by LLMs. Unlike previous benchmarks, EffiBench-X covers six programming languages (Python, Java, JavaScript, Ruby, C++, and Go), includes complex tasks drawn from competitive programming platforms, and uses human-expert solutions as baselines. The paper evaluates 26 LLMs using metrics such as Execution Time (ET), Memory Peak (MP), and Memory Integral (MI), and reveals a large gap between human and LLM-generated code efficiency.

**Dataset Code Accessibility:**

Yes

**Ethical Considerations:**

No, there are no or only very minor ethics concerns

**Final Justification:**

I appreciate the author's response in the rebuttal, and raised my score.

**Limitations Weaknesses:**

While the benchmark is comprehensive, the paper does little to extract scientific understanding from it. The conclusions largely restate numbers without deeper analysis of why LLMs produce inefficient code. There is no examination of: Common inefficiencies (e.g., redundant loops, poor algorithm choice). Whether inefficiencies are semantic (algorithmic) or syntactic (language/library use). How this varies by language.

EffiBench-X currently benchmarks single-shot completions from LLMs in isolation. But in practice: Code generation often happens in interactive loops (e.g., agent retries based on failures, or adapts to performance).

Happy to raise my score if the questions are answered

(i've raised the score, thanks for the rebuttal)

**Strengths Contributions:**

First benchmark to systematically evaluate efficiency of LLM-generated code across multiple languages. This addresses a crucial gap in current evaluation ecosystems focused mainly on correctness or Python-only efficiency.

Diverse, efficiency-critical problems are collected from competitive programming platforms such as Codeforces, AtCoder, and LeetCode, with an explicit filtering to avoid data contamination.
Canonical human-expert solutions serve as gold standards, validated via real submissions and rigorous correctness checks.

Identifies language-specific differences in performance, showing models perform better in dynamic languages like Python and Ruby than statically-typed ones like Java and C++.

---

> ### Author Rebuttal · Authors · 2025-07-31
>
> We would like to extend our sincere gratitude to the reviewer for the insightful and valuable comments. We have provided a detailed, point-by-point response below to address each of the concerns raised.
>
> We believe our clarifications and additional experiments have substantially improved the manuscript. We would be very grateful if you would consider increasing the overall score if you find our responses have satisfactorily addressed your concerns.
>
> **W1. Lack of Deeper Scientific Analysis**
>
> Thanks for the great suggestion! We've since performed a deeper analysis to understand *why* LLMs produce inefficient code and looked at how these issues differ across programming languages. Our analysis shows the problems aren't just minor mistakes; they're often fundamental flaws in logic. We found the issues generally fall into a few camps: choosing a bad algorithm, performing redundant work, using the wrong data structure, poor implementation choices, or slow I/O. Ultimately, most inefficiencies are **semantic**—a flaw in the model's problem-solving logic, rather than purely **syntactic**. The biggest culprits across the board are **Algorithmic Complexity Deficiency**, **Redundant Computation**, and **Suboptimal Implementation**. This means the models tend to understand the goal but choose a naive, brute-force path to get there.
>
> A perfect example is LeetCode problem 3422. The LLM's solution correctly simulates the problem second-by-second, but it's slow, running in $O(n \\cdot k)$ time.
>
> ```python
> # Inefficient LLM Solution (Simulation)
> class Solution:
>     def valueAfterKSeconds(self, n: int, k: int) -> int:
>         MOD = 10**9 + 7
>         a = [1] * n
>         for _ in range(k):
>             # This inner loop makes the whole thing slow
>             for i in range(1, n):
>                 a[i] = (a[i] + a[i-1]) % MOD
>         return a[n-1]
> ```
>
> The optimal solution, however, uses a mathematical insight (combinations) to solve it much faster. This shows the model's failure to make deeper, abstract connections.
>
> ```python
> # Efficient Canonical Solution (Math Insight)
> class Solution:
>     def valueAfterKSeconds(self, n: int, k: int) -> int:
>         import math
>         return math.comb(k + n - 1, k) % 1_000_000_007
> ```
>
> **So does Language Make a Difference?** For the most part, the core problems are universal. **Algorithmic Complexity Deficiency** is a top issue in every language we tested. This suggests the weakness is in the LLM's fundamental reasoning, not its grasp of a specific language's syntax.  However, we did see some interesting language-specific trends:
>
> In dynamic languages like **Python and Ruby**, **Redundant Computation** and **Suboptimal Implementation** are especially common. Their flexible syntax makes it easy to write code that works but secretly does a lot of extra work, a trap the LLMs fall into often. For **Java and JavaScript**, **I/O and Language-Level Overhead** was a bigger issue (\~15% of problems). This often involved using unbuffered I/O or other slow standard library functions that are common but not performant. In **C++**, where performance is critical, we saw a higher percentage of **Inefficient Data Structure** choices (14.5%). Picking the wrong container in C++ has a huge impact, and it's a nuance the models frequently miss.
>
> This deeper analysis has been crucial for the paper, and we thank you for pushing us to look more closely at the results. It has allowed us to provide a much clearer scientific understanding of how and why LLMs fail at generating efficient code.
>
>
>
> **W2. Focus on Single-Shot Generation**
>
> We thank the reviewer for their insightful comment. We agree that exploring more sophisticated prompting strategies and iterative refinement are crucial for advancing the practical capabilities of LLMs in code generation. Following this valuable suggestion, we have conducted a new set of experiments to investigate these dimensions, which we will add to the paper. Our new experiments confirm that both advanced prompting for single-shot generation and iterative refinement based on performance feedback substantially improve the model's ability to produce efficient code. These results underscore the potential for LLMs to function as adaptive agents that respond to detailed, contextual instructions.
>
> Our first set of experiments focused on the **impact of prompt design in single-shot generation**. We evaluated several advanced prompts designed to provide richer context, including a persona (`aggressive_opt`), hardware environment details (`hardware_hint`), and explicit optimization goals (`ecco`, `eco_aware`).
>
>
> - **Efficiency-Aware Prompt (ecco)**: Explicitly requests code that is efficient in both runtime and memory, based on the ECCO framework.
> - **Hardware-Hinting Prompt (hardware_hint)**: Informs the model about the target execution environment (e.g., Linux, x86-64 CPU, g++ -O2), as suggested.
> - **Aggressively Optimized Prompt (aggressive_opt)**: Demands aggressive and specific optimization for both runtime and memory.
> - **Eco-Aware Prompt (eco_aware)**: Frames the optimization goal in terms of minimizing CPU cycles and energy consumption, assuming a power-sensitive environment.
>
> As shown in Table 15, all advanced prompts significantly outperformed our baseline with GPT-4o-mini. This demonstrates that providing more detailed instructions—whether about the execution environment, optimization targets, or a specific persona—markedly enhances code generation quality. Notably, the `eco_aware` prompt, which frames efficiency in terms of minimizing resource consumption, yielded the best overall performance, achieving a runtime pass rate of $32.40\%$ and a memory pass rate of $32.20\%$.
>
> *Table 15: Performance of Single-Shot Generation with Advanced Prompts*
>
> | Prompt Strategy | Runtime Pass Rate | Memory Pass Rate | Integral Score |
> | :--- | :---: | :---: | :---: |
> | Baseline Prompt | 27.12% | 36.21% | 28.26% |
> | `aggressive_opt` | 31.36% | 31.34% | 31.24% |
> | `ecco` | 31.99% | 31.49% | 31.73% |
> | `hardware_hint` | 30.64% | 30.52% | 30.63% |
> | `eco_aware` | **32.40%** | **32.20%** | **32.42%** |
>
> In our second set of experiments, we investigated **iterative refinement by providing the LLM with performance feedback** on its generated code. We adopt **EffiLearner** [1] as our primary approach, which is the state-of-the-art method on Effi-Bench [2]. EffiLearner supplied the model with comprehensive reports, including total runtime, peak memory, and detailed line-by-line profiler data. We found that the quality of feedback is paramount. The `Unsupervised Self-Refine` baseline [3], which lacked any performance data, showed negligible improvement. In contrast, providing rich, granular feedback proved highly effective. Both **EffiLearner** and the `Memory Profiler` strategy, which included line-by-line memory usage, achieved the highest scores, with **EffiLearner** reaching a peak memory pass rate of $40.59\%$. This confirms that LLMs benefit most from understanding precisely *where* inefficiencies exist, not just that they are present.
>
> *Table 16: Performance of Iterative Refinement with Different Feedback Types*
>
> | Refinement Strategy | Runtime Pass Rate | Memory Pass Rate | Integral Score |
> | :--- | :---: | :---: | :---: |
> | Unsupervised Self-Refine | 34.39% | 36.63% | 34.17% |
> | Result Aware | 37.16% | 37.93% | 36.72% |
> | Line Profiler | 35.38% | 36.64% | 35.02% |
> | Memory Profiler | 39.15% | 39.92% | 38.85% |
> | **EffiLearner (Full Feedback)** | **39.19%** | **40.59%** | **38.83%** |
>
> Prompts used in Table 16 follow the format of the EffiLearner GitHub Repo.
>
> References:
>
> [1] EffiLearner: Enhancing Efficiency of Generated Code via Self-Optimization. NeurIPS 2024
>
> [2] EffiBench: Benchmarking the Efficiency of Automatically Generated Code. NeurIPS 2024
>
> [3] Reflexion: Language agents with verbal reinforcement learning. NeurIPS 2023

---

### Official Review · Reviewer_MDVc · 2025-07-03

**Rating:** 4
**Confidence:** 3

**Summary:**

This paper introduces Effibench-X, a multi-lingual benchmark focused on measuring the runtime efficiency of code generated by LLMs. It contains competitive programming problems across six programming languages, paired with solutions either from human experts or translated from expert solutions in another language. The paper uses a Docker-based sandbox environment with high-resolution profiling to ensure reliable evaluation. The performance differences between open-source and closed models, models of different sizes and capabilities are compared, showing that LLMs are generally less efficient in generating code, with Qwen3-32B achieving the closest to human performance (62%).

**Dataset Code Accessibility:**

Yes

**Dataset Code Comments:**

The Huggingface datasets implementation is easy to load and use, and the GitHub code is well documented.

**Ethical Considerations:**

No, there are no or only very minor ethics concerns

**Final Justification:**

The rebuttal addressed my main question about sampling strategy and the comparison against CodeScope. Therefore, I decided to update my score to 4.

**Limitations Weaknesses:**

- There has been [prior work](https://arxiv.org/abs/2311.08588) (CodeScope) that is also multi-lingual and covers multiple tasks, one of which is efficiency. This paper did not cite or discuss CodeScope, which weakens the paper's novelty as this paper claims to be the first benchmark that measures the efficiency of LLM-generated code.
- All of the experiments only have a pass@1 number, are all of the LLM solutions generated via greedy decoding? How would taking multiple samples with a non-zero temperature affect the efficiency of the solutions?
- It's not quite clear how the human expert solution is being selected. If a problem has multiple solutions, is the best one picked as the expert solution? Is it representative enough of the average human performance?

**Strengths Contributions:**

- Generating efficient programs is as important as generating correct programs, therefore the benchmark targets an important problem.
- Well-designed sandboxed execution environment to minimize variance in profiling
- This benchmark introduces a multilingual dataset, whereas most of the previous benchmarks only covers a single language like C++ or Python

---

> ### Author Rebuttal · Authors · 2025-07-31
>
> We would like to extend our sincere gratitude to the reviewer for the insightful and valuable comments. We have provided a detailed, point-by-point response below to address each of the concerns raised.
>
> **W1: Missing Citation of Prior Work**
>
> We thank the reviewer for this valuable comment and for pointing out the omission of CodeScope. We will incorporate a thorough discussion of CodeScope into our manuscript in the next version and provide a clear comparison to situate the contributions of EffiBench-X. Accordingly, we will revise our novelty claim to more accurately reflect our contribution in the context of prior art.
>
> Specifically, we will highlight the following critical distinctions that underscore the novelty and necessity of EffiBench-X:
>
>
>
> * **Benchmark Scale and Comprehensiveness:** Another limitation of CodeScope is its scale. With only 30 tasks per language for code efficiency measurement, it can offer an initial snapshot but may lack the statistical power for a robust and reliable evaluation. In contrast, EffiBench-X features an order of magnitude more problems, with over 600 tasks for each supported language. This extensive scale is crucial for ensuring that the evaluation is comprehensive, reduces variance, and provides a reliable measure of a model's true code generation capabilities rather than its performance on a small, specific subset of tasks.
>
> * **Language Coverage and Relevance:** CodeScope focuses on Python and C++. While important, this overlooks other major languages used extensively in software development. EffiBench-X provides significantly broader coverage by including not only Python and C++ but also **Java**, **Go**, **Ruby**, and **JavaScript**. The inclusion of a language like Java is critical, given its persistent top ranking in industry usage (e.g., TIOBE Index [1]). The other languages were chosen to represent diverse paradigms, from systems programming (Go) to web development (Ruby, JS), offering a more holistic assessment of a model's multi-lingual proficiency.
>
> * **Data Contamination:** The tasks in CodeScope were collected from programming contests that occurred before September 2023. As this timeframe precedes the knowledge cutoffs for the most powerful LLMs, there is a high probability of data leakage. This raises significant concerns about the validity of its results, as high scores could be attributed to a model's ability to memorize solutions seen during training, rather than its ability to generate efficient code for new problems. Unlike CodeScope, EffiBench-X was constructed exclusively from problems and data sources created after these widely recognized knowledge cutoffs. This deliberate design choice is a cornerstone of our contribution, ensuring that our benchmark provides a true and untainted measure of a model's code intelligence.
>
> In our revision, we will reposition our contribution not as the absolute first to consider efficiency, but as the **first large-scale, multi-lingual benchmark specifically designed for robust and comprehensive efficiency evaluation of LLM-generated code**. We believe this clarification, along with the detailed comparison, will properly contextualize our work and highlight its significant advancements over the existing literature.
>
> **W2. Limited to Greedy Decoding (pass@1)**
>
> We thank the reviewer for their insightful comment regarding the evaluation being limited to greedy decoding (pass@1). This is a valid point, and we have conducted additional experiments to investigate how stochastic sampling affects the efficiency of the generated solutions.
>
> To address this, we followed the reviewer's suggestion and adopted a multi-sampling strategy. For a representative set of models (including GPT-4o, Claude 3.5 Sonnet, Qwen-32B, and DeepSeek-V2), we generated five solutions (n=5) for each problem using a temperature of 0.8. From these generated samples, we evaluated the results using two approaches:
>
> * **Most Efficient (pass@5)**: For each problem with at least one correct solution among the five samples, we select the single most efficient correct solution for scoring. This measures the best potential efficiency an LLM can achieve with multiple attempts.
>
> * **Average Efficiency (pass@5)**: For each problem, we calculate the average efficiency score across all functionally correct solutions within the five samples. This provides a more conservative measure of the expected efficiency.
>
> *Table 13: Most Efficient Solution Scores (pass@5)*
>
> | Model | Runtime Score | Memory Score | Integral Score | Pass Rate |
> | :--- | :--- | :--- | :--- | :--- |
> | **DeepSeek-V2** | 41.80% | 44.49% | 43.04% | 46.07% |
> | **Qwen-32B** | 27.64% | 29.13% | 28.61% | 29.86% |
> | **GPT-4o** | 26.15% | 28.09% | 26.89% | 28.57% |
> | **Claude 3.5 Sonnet** | 31.88% | 34.83% | 32.97% | 35.47% |
>
>
> *Table 14: Average Efficiency Scores of Correct Solutions (pass@5)*
>
> | Model | Average Runtime Score | Average Memory Score | Average Integral Score |
> | :--- | :--- | :--- | :--- |
> | **DeepSeek-V2** | 34.88% (34.45%, 35.68%) | 35.76% (35.24%, 36.36%) | 34.28% (33.27%, 35.05%) |
> | **Qwen-32B** | 22.03% (21.14%, 22.80%) | 22.80% (22.20%, 23.33%) | 21.93% (20.83%, 22.68%) |
> | **GPT-4o** | 46.89% (44.45%, 48.25%) | 47.74% (45.37%, 49.37%) | 46.88% (44.81%, 48.41%) |
> | **Claude 3.5 Sonnet**| 28.28% (27.32%, 28.86%) | 29.09% (28.46%, 29.59%) | 28.16% (27.54%, 28.63%) |
>
> Our analysis shows that stochastic sampling improves both functional correctness (pass rate) and, consequently, the final efficiency scores compared to the original greedy decoding results. For instance, selecting the most efficient sample (pass@5) boosts the integral efficiency scores by a few percentage points across the board. However, we observe that the overall performance gap between LLM-generated solutions and human-written code remains substantial. Furthermore, the relative efficiency ranking among the different models is largely preserved, reinforcing our original conclusions.
>
> We appreciate the reviewer's suggestion, as this additional analysis provides a more nuanced view of LLM performance and strengthens our findings. We have incorporated these results and discussion into the revised manuscript.
>
>
> **W3. Unclear Human Expert Solution Selection**
>
>
> When multiple human-written solutions exist for a single problem, we follow existing works (e.g., EffiBench and ENAMEL) and select the one with the highest efficiency (i.e., the lowest execution time) as the definitive expert solution.
>
> The key reason for this choice is that our benchmark does not aim to represent average human performance. Instead, the expert solution is intended to function as a gold standard or an optimal reference point. The goal is to establish a high-quality, efficient target that code LLMs should strive to meet or exceed. Using the most performant human solution provides a clear, objective, and ambitious target for evaluation. An "average" solution would be less informative for pushing the boundaries of performance and is inherently difficult to define robustly.

---

> > ### Comment · Reviewer_MDVc · 2025-08-06
> >
> > Thank you to the authors for taking the time to respond to my comments and address all questions raised in my review. I appreciate the additional abalation on sampling strategy and additional discussion on how the paper is positioned. Overall I'm happy with the response and will adjust my score accordingly.

---

> > > ### Author Response · Authors · 2025-08-07
> > >
> > > Thank you for recognizing our work. Your feedback is valuable to us as we enhance it.

---

### Official Review · Reviewer_RKmc · 2025-07-13

**Rating:** 5
**Confidence:** 5

**Summary:**

The authors introduce EFFIBENCH-X, a new benchmark that measures how efficiently LLMs generate code across six programming languages. The suite contains 623 recently released competitive-programming problems, each bundled with a human-expert “canonical” solution, an automatically validated test-case generator, and a language-specific test harness. Efficiency is reported along three metrics.  Studying across a large number of open- and closed-source models shows that even the best model attains only about 62% of human execution-time efficiency when averaged over all tasks.

**Additional Feedback:**

Few questions/comments:

1- It would be great to identify tasks where ET is clipped at 1.0, inspect complexity classes or the presence of advanced data structures, and sample whether the speed-up stems from an algorithmic shift. Even a manual audit of, say, the 50 most accelerated solutions would be a good start.

2- I still think a data contamination analysis would be useful, esp. since this is one of your core contribution.

3- The triple-run experiment is a good start, but expanding it to other CPUs would clarify whether the current setup truly “eliminates” variability.

4- Consider adding the median speed-ratio (T_human / T_model) and the proportion of tasks where that ratio exceeds 1.

5- Did you observe any tasks where all models surpassed the human solution? Collective super-human performance might signal that the canonical code is not actually optimal.

6- Execution-time limits are fixed at 10s. Have you profiled whether a tighter cap changes relative rankings?

7- Gemini-2.5-Pro shows strikingly high memory-peak efficiency yet mediocre execution time. Is this due to language choices (e.g., Python versus Java) or inherent model characteristics?

8- Breaking results down by algorithm category (dynamic programming, graph shortest paths, number theory, etc.) might highlight strengths and weaknesses in current LLM reasoning.

9- Have you tried hinting the target hardware in your prompt? Have you tried different prompting to see if the models can possibly navigate the code optimization landscape better?

**Dataset Code Accessibility:**

Yes

**Ethical Considerations:**

No, there are no or only very minor ethics concerns

**Final Justification:**

Already left a comment.

**Limitations Weaknesses:**

(1) Without an empirical overlap analysis against common pre-training corpora, readers cannot gauge how much leakage remains. A simple n-gram or embedding-based search over repositories would strengthen the claim.

(2) The paper explains how generators are validated, but never reports how often they fail to compile, produce too few cases, or exercise only trivial paths. Publishing the generator pass-rate, average branch or statement coverage (perhaps estimated on canonical solutions), and typical repair cost would let future users judge robustness (I have not checked the appendix).

(3) ET, MP and MI are clipped at 1.0, so a solution that is twice as fast as the expert scores the same as one that merely ties. This design masks important progress. A complementary statistic, such as the median ratio of human time to model time and the fraction of tasks where the model is faster, could be reported without abandoning the current scale.

(4) Inter-machine variation and language-specific jitter remain unexplored. Even a small cross-hardware study would reassure practitioners who intend to rerun the suite on different infrastructure.

(5) The results section offers qualitative anecdotes but no systematic look at whether models choose inferior algorithms, mishandle I/O, or simply neglect micro-optimizations. Lightweight static analyses, such as loop nest depth, asymptotic complexity heuristics, or library-call profiling, could reveal where optimization effort should focus.

(6) EFFIBENCH-X does not cover systems-level efficiency factors such as vectorization, cache locality, or asynchronous I/O. The authors already acknowledge this; outlining a concrete roadmap for adding real-world workloads would increase the benchmark’s long-term impact.

**Strengths Contributions:**

- EFFIBENCH-X fills the gap in the landscape of code efficiency benchmarks. Previous efficiency suites have been either mono-lingual or restricted to small subsets of HumanEval, MBPP, or LeetCode. The authors make an attempt to curb data contamination. The benchmark is not only larger than earlier multi-language sets but also demonstrably harder: most leading models still fail more than a third of tasks, and when they succeed they remain markedly slower or more memory-hungry than the best human baseline.

- The evaluation pipeline itself is carefully engineered. Asking an LLM to write a test-case generator rather than hundreds of individual cases is an interesting approach. A multi-stage validation loop repairs faulty templates automatically and flags hard cases for manual vetting, which inspires confidence in the correctness metric. The sandbox shows low run-to-run variance.

- Finally, the motivation of the paper is clear, related work is decent, and figures of representative “slow” versus “fast” code snippets help the reader see where models waste resources. The appendix spells out the exact container images, compiler flags, profiler details, and model prompts, making reproducibility easier.

---

> ### Author Rebuttal · Authors · 2025-07-31
>
> We would like to extend our sincere gratitude to the reviewer for the insightful and valuable comments. We have provided a detailed response below to address each of the concerns raised.
>
> **W1&A2. Lack of Data Contamination Analysis**
>
> Following the practices suggested in recent literature [1], we analyzed our 623 tasks using n-gram overlap and embedding-based searches. The analysis, summarized in Table 5, confirms minimal contamination. Our most stringent method, an embedding search, identified only 3 contaminated tasks (0.48%).
>
> *TABLE 5: Contamination Analysis*
>
> | Method | Parameters | Contaminated Tasks | Rate (%) |
> | :--- | :--- | :--- | :--- |
> | **N-gram Overlap** | n=20 | 11 / 623 | 1.77 |
> | **N-gram Overlap** | n=30 | 6 / 623 | 0.96 |
> | **Embedding Search**| Cosine Similarity > 0.90 | 3 / 623 | 0.48 |
>
> Ref:
>
> [1] Key-Point-Driven Data Synthesis with its Enhancement on Mathematical Reasoning. Arxiv 2025
>
> **W2. Missing Test-Case Generator Metrics**
>
> To evaluate our test-case generator, we measured its generation success rate and code coverage. As shown in Table 6, our generator is highly effective. It succeeds on the first attempt 81.86% of the time, reaching a 96.47% success rate within three automated retries. Furthermore, the generated tests are comprehensive, achieving 100% line and branch coverage against the canonical solutions (Table 7).
>
> *TABLE 6: Success rate of test generator*
>
> | Attempts | # of Generators | Ratio | Cumulative Ratio |
> | :---: | :---: | :---: | :---: |
> | 1 | 510 | 81.86% | 81.86% |
> | 2 | 71 | 11.40% | 93.26% |
> | 3 | 20 | 3.21% | 96.47% |
> | 4 | 4 | 0.64% | 97.11% |
> | 5 | 3 | 0.48% | 97.59% |
> | >5 | 15 | 2.41% | 100.00% |
>
> *TABLE 7: Coverage analysis*
>
> | Metric | Results|
> | :--- | :---: |
> | Line Coverage | $100.0%$ |
> | Branch Coverage | $100.0%$ |
>
>
> **W3&A1&A4. Clipped Efficiency Metrics**
>
> We re-analyzed performance with unclipped ET ratios, which revealed that models like Claude 3.7 Sonnet produce faster-than-expert code over 20% of the time (Table 8). Crucially, the overall relative model rankings remained consistent, confirming our original conclusions. To understand the source of these gains, we categorized the optimization techniques and discovered distinct model strategies (Table 9). While Gemini 2.5 Pro and Claude 3.7 Sonnet typically apply implementation-level optimizations (e.g., achieving a 3.33x speedup by avoiding str.swapcase() overhead in a loop), GPT-4o's advantages stem from high-level reasoning, such as improving algorithmic complexity (e.g., O($N^2$) to O($NlogN$)) or complete problem reformulation.
>
> *TABLE 8: Unclipped median ET ratio*
>
> | Model | Median Unclipped ET Ratio | % of Tasks with ET > 1.0 |
> | :--- | :---: | :---: |
> | GPT-4o-mini | 0.77 | 1.61% |
> | GPT-4o | 0.82 | 7.54% |
> | Claude 3.5 Haiku | 0.91 | 9.15% |
> | Gemini 2.5 Flash | 0.87 | 13.80% |
> | Gemini 2.5 Pro | 0.91 | 19.26% |
> | Claude 3.7 Sonnet | 0.97 | 20.71% |
>
> *TABLE 9: Faster code analysis*
>
> | Optimization Category | Gemini 2.5 Pro | Claude 3.7 Sonnet | GPT-4o |
> | :--- | :---: | :---: | :---: |
> | Implementation-Level Optimization | **55.83%** | **62.02%** | 13.79% |
> | Algorithmic Complexity Optimization | 26.67% | 20.93% | 37.93% |
> | Problem Reformulation | 7.50% | 6.98% | **41.38%** |
> | Advanced Data Structure Usage | 5.00% | 3.88% | 0.00% |
> | Pruning and Heuristic Optimization | 5.00% | 6.20% | 6.90% |
>
> ```python
> ## LLM Solution (Claude 3.7 Sonnet)
> def get_kth_char(s, k):
>     original_len = len(s)
>     k -= 1 # 0-indexed
>     pos = k % original_len
>     # Count set bits to determine if case should be flipped
>     flip = bin(k // original_len).count('1') % 2 == 1
>     char = s[pos]
>     if flip:
>         return char.upper() if char.islower() else char.lower()
>     return char
> # ... main solve function ...
>
> ## Canonical Solution
> s = input()
> q = int(input())
> n = len(s)
> a = []
> for k in input().split():
>     k = int(k) - 1
>     if (k // n).bit_count() % 2 == 0:
>         a.append(s[k % n])
>     else:
>         a.append(s[k % n].swapcase()) # Slower due to new string creation
> print(*a)
>
> ```
>
> **W4&A3. Hardware Variation**
>
> We conducted experiments on different hardware to validate benchmark's robustness. Our findings confirm that the results are hardware-agnostic. The relative efficiency ranking of the models (Gemini-2.5-Pro > DeepSeek-R1 > GPT-4o) was identical on both AMD and Intel systems. Furthermore, our normalization method effectively abstracts hardware differences, as the final integral scores for each model varied by less than 3% between the platforms. This validates our methodology as a reliable and generalizable measure of model efficiency.
>
> *TABLE 10: Different hardware*
>
> | Hardware         | Model          | Runtime | Memory  | Integral |
> | :--------------- | :------------- | :------ | :------ | :------- |
> | **AMD EPYC** | Gemini-2.5-Pro | 56.83%  | 57.45%  | 54.13%|
> | (m7a)            | DeepSeek-R1    | 48.92%  | 49.44%  | 47.03%|
> |                  | GPT-4o         | 39.04%  | 39.00%  | 38.59%|
> | **Intel Xeon** | Gemini-2.5-Pro | 57.00%  | 58.54%  | 54.86% |
> | (m5.metal)       | DeepSeek-R1    | 52.08%  | 53.06%  | 49.93% |
> |                  | GPT-4o         | 39.46%  | 40.80%  | 38.93% |
>
>
> **W5. Lack of Systematic Inefficiency Analysis**
>
> We've performed a systematic analysis of inefficiencies in code generated by Gemini 2.5 Pro, GPT-4o, and Claude 3.7 Sonnet in **Review rddN Table 3**. Our analysis reveals three dominant error types: Algorithmic Complexity Deficiency, Redundant Computation, and Suboptimal Implementation. For example, models frequently generate brute-force $O(n^2)$ solutions instead of optimal $O(n \log n)$ ones or use inefficient data structures like a Python `list` for a queue instead of `collections.deque`. Crucially, severe performance failures (efficiency < 10%) are almost exclusively caused by these fundamental **algorithmic deficiencies**. This indicates that future work should prioritize improving the models' **high-level algorithmic reasoning** rather than focusing on low-level micro-optimizations.
>
>
> **W6. Limited Scope of Efficiency Factors**
>
> We thank the reviewer for this excellent suggestion. We will expand EffiBench-X to incorporate systems-level efficiency. Our plan involves:
> 1.  **New Benchmarks:** Introducing tasks from HPC and I/O-intensive domains to test for vectorization, cache locality, and asynchronous I/O.
> 2.  **Enhanced Evaluation:** Capturing granular system metrics like cache misses and I/O throughput using profiling tools.
> 3.  **New Baselines:** Developing expert-written solutions to set a gold standard for performance.
>
> **A5. Investigate Super-Human Performance Cases**
>
> After conducting performance analysis, we do not find any task where the solutions generated by all LLMs are better than the expert solution.
>
> **A6. Profile with Tighter Time Limits**
>
> We re-evaluated all models with the execution time limit halved from 10 seconds to 5 seconds. The results show that performance is remarkably stable, with negligible changes in scores for all models. For instance, Claude 3.7 Sonnet's score remains identical at 10s, 7s, and 5s, with only a minor drop at a very aggressive 3s limit.  Most importantly, the relative performance ranking of all models remains completely unchanged with the 5-second cutoff. This confirms that our original 10-second time limit is a robust choice that does not unduly favor slower models, and that most successful code generations are completed well within this window.
>
>
> *Table 11. Performance of Claude 3.7 Sonnet at Various Cutoffs*
> | Time Limit | ET (%) |
> | :--- | :--- |
> | 10s (Original) | 53.63 |
> | 7s | 53.63 |
> | 5s | 53.63 |
> | 3s | 53.50 |
>
> *Table 12. Ranking of LLMs on different timeout setups*
>
> | Model | ET @ 10s Cutoff (%) | ET @ 5s Cutoff (%) | Change (%) |
> | :--- | :--- | :--- | :--- |
> | Gemini 1.5 Pro | 64.92 | 64.92 | 0.00 |
> | Gemini 1.5 Flash | 58.95 | 58.88 | -0.07 |
> | Claude 3.7 Sonnet | 53.63 | 53.63 | 0.00 |
> | GPT-4o | 40.12 | 40.12 | 0.00 |
> | Claude 3 Haiku | 35.97 | 35.95 | -0.02 |
> | GPT-4o mini | 28.63 | 28.62 | -0.01 |
>
>
> **A7. Analyze Gemini's Performance Anomaly**
>
> The performance of Gemini-2.5-Pro is an inherent characteristic of the code it generates rather than an artifact of specific programming languages. The evaluation results presented are averages, and it's crucial to note that this performance profile is consistent across all six programming languages tested in EffiBench-X.It results from the fundamental **space-time complexity trade-off** in algorithm design. The model excels at creating space-optimized code, leading to its state-of-the-art memory performance. However, prioritizing minimal memory usage often results in less competitive execution times.
>
> **A8. Breakdown Results by Algorithm Category**
>
> We analyzed performance by algorithm category (see **Rebuttal Review rddN Table 4**). The model excels at procedural tasks like string-matching (90.91%) but struggles with those requiring abstract reasoning, such as dynamic programming (23.09%) and greedy algorithms. This analysis will be incorporated into the next version of the paper.
>
>
> **A9. Explore Advanced Prompting Strategies**
>
> We conducted new experiments that will be added to the paper. As shown in our response to **Review 2u2z Tables 15 and 16**, these results demonstrate that advanced prompting improves LLM-generated code efficiency. However, a gap remains when compared to human-expert solutions.

---

> > ### Author Response · Authors · 2025-08-05
> > **Friendly Reminder**
> >
> > We sincerely appreciate the time and effort you have dedicated to reviewing our work. In response to your valuable feedback, we have provided detailed explanations for the issues raised.
> >
> > As the discussion period progresses, we are eager to hear your thoughts on our responses, including whether they have adequately addressed your concerns. If our revisions and discussions indicate the potential for a score adjustment, we would be very grateful for your consideration.
> >
> > We are committed to incorporating all of your suggestions to further enhance the quality of our manuscript. We look forward to your further comments and discussion.

---

### Official Review · Reviewer_rddN · 2025-07-19

**Ethics Flags:** Discrimination, bias, and fairness
**Rating:** 5
**Confidence:** 4

**Summary:**

The paper proposes the first multi-language benchmark, EFFIBENCH-X, designed to measure the efficiency of LLM-generated code. EFFIBENCH-X supports evaluations for Python, C++, Java, JavaScript, Ruby, and Golang, and involves competitive programming tasks with human-expert solutions as efficiency baselines. Extensive evaluations on SOtA LLMs show that these models generate functionally correct code, but they consistently underperform human experts in efficiency.

**Additional Feedback:**

1. The authors should provide more structural details about the collected cases and introduce the instruction templates for different LLMs.
2. The authors could utilize a more diversified presentation to show and reveal the evaluation results.

**Dataset Code Accessibility:**

Partly

**Dataset Code Comments:**

The data structure of the evaluation dataset is not clarified. Especially since the instructions are not specified for different LLMs, the generation results of each LLM may not be reproducible.

**Ethical Comments:**

See the above comments.

**Ethical Considerations:**

Yes, there are ethics concerns that require attention by the authors

**Final Justification:**

The authors have addressed my concerns and also promised to fix the issues in the final version. I raised the score to acc.

**Limitations Weaknesses:**

1. The structure of the evaluation cases is not provided or summarized. A detailed global description of the evaluation dataset would help understand the reliability of the evaluation.
2. The instructions for different LLMs to generate corresponding codes are not introduced in detail. Different instruction templates may seriously impact the quality and even efficiency of the generated codes of different LLMs. This would greatly influence the reliability of the reported results.
3. Using tables to present results is overly monotonous. Most results are reported in tables. The experimental section lacks a more comprehensive presentation to reveal more interesting conclusions.

**Strengths Contributions:**

1. The motivation and contribution of the benchmark paper are clear. The proposed benchmark facilitates the evaluation of the correctness and efficiency of multi-language LLM-generated codes.
2. Extensive results on various settings verify the performance of various proprietary LLMs and reveal interesting findings.
3. The paper is well-written and easy to follow. The paper's presentation is clean and informative.

---

> ### Author Rebuttal · Authors · 2025-07-31
>
> We would like to extend our sincere gratitude to the reviewer for the insightful and valuable comments. We have provided a detailed, point-by-point response below to address each of the concerns raised.
>
>
> **W1&A1. Lack of Dataset Structure Details**
>
> We sincerely thank the reviewer for their constructive feedback. Based on your suggestion, we provide a detailed breakdown of our dataset's structure and the instruction template used for querying the LLMs. We will incorporate these details into the appendix of our revised manuscript to improve its clarity and reproducibility. First, to enhance clarity on the structure and composition of EffiBench-X, we present a detailed dataset card in Table 1 and a summary of the task distribution by source.
>
> *TABLE 1: Dataset Card of EffiBench‑X*
>
> | Field Name | Definition |
> | :--- | :--- |
> | `id` | Problem index in the corresponding source. |
> | `title` | Task name. |
> | `title_slug` | URL‑friendly slug of the task name, used for referencing. |
> | `description` | Task description. |
> | `description_md` | Task description in markdown version. |
> | `source` | The source platform from which the problem was collected. |
> | `url` | URL to the original problem. |
> | `type` | Problem type: functional or I/O. |
> | `starter_code` | Starter code provided in multiple programming languages (C++, Java, Python, JavaScript, Go, Ruby). |
> | `solutions` | Canonical solutions for each language. For each language, `runtime` represents the solution optimized for minimal runtime (with reasonably low memory), and `memory` represents the solution optimized for minimal memory (with reasonably low runtime). |
> | `test_case_generator` | The test case generator is used to produce evaluation test cases. |
> | `generated_tests` | Tests generated by the `test_case_generator`. These tests are used in our experiments. |
> | `test_runner` | Test templates for functional problems. For each target language, a code template is generated to create a runnable program. It contains a special placeholder for injecting the solution and handles input deserialization, function invocation and output serialization. |
>
> Next, to offer a global perspective on our dataset, we provide the distribution of EffiBench-X across five major competitive programming websites in Table 2. This diversity ensures that EffiBench-X covers a wide range of problem styles, complexities, and constraints, contributing to a robust and reliable evaluation.
>
> *TABLE 2: Task Distribution by Source Websites in EffiBench‑X*
>
> | Website | Aizu | CodeChef | CodeForces | AtCoder | LeetCode | Total |
> | :--- | :--- | :--- | :--- | :--- | :--- | :--- |
> | **Tasks** | 33 | 93 | 10 | 149 | 338 | **623** |
>
>
>
>
> **W2&A1. Missing Instruction Templates**
>
> We thank the reviewer for raising this important point. Our evaluation follows existing works (e.g., EffiBench and Mercury) and uses a single, unified instruction template for all LLMs. This is a standard practice in comparative LLM evaluation and is crucial for ensuring a fair and controlled comparison. Using model-specific instruction templates would introduce a confounding variable (i.e., prompt engineering), making it impossible to isolate the models' intrinsic capabilities. This standardized format directly addresses the concern regarding **fairness and bias**, as all models are subjected to the same conditions, eliminating any potential for researcher-induced bias through prompt optimization.
>
> The uniform prompt template used is:
>
> ```
> You are a competitive programming expert skilled in multiple programming languages.
> Your task is to solve the following problem in {target_lang}:
>
> {description}
> ```
>
> **W3 & A2: The experimental section lacks a more comprehensive presentation to reveal more interesting conclusions.**
>
> We thank the reviewer for this suggestion. We have conducted new analyses to provide a more comprehensive experimental section. Due to the rebuttal platform's limitations, we present these findings in tables for now, but will incorporate richer visualizations like Radar Charts and minted figures in the final paper. To understand why some models generate slower code, we analyzed the causes of inefficient code. The primary issues were Algorithmic Complexity Deficiency, Suboptimal Implementation, and Redundant Computation. Crucially, the most severe performance failures are almost exclusively caused by fundamental Algorithmic Complexity deficiencies, not micro-optimizations.
>
> *TABLE 3: Inefficiency Analysis*
>
> | Inefficiency Category | Gemini 2.5 Pro | GPT-4o | Claude 3.7 Sonnet |
> | :--- | :---: | :---: | :---: |
> | Algorithmic Complexity | **29.9%** | **37.1%** | 28.0% |
> | Suboptimal Implementation | 28.2% | 16.3% | **29.0%** |
> | Redundant Computation | 22.9% | 22.9% | **29.5%** |
> | I/O Overhead | 11.4% | 17.5% | 5.3% |
> | Inefficient Data Structure | 7.6% | 6.2% | 8.2% |
>
> For instance, an inefficient simulation for leetcode_3422_find-the-n-th-value-after-k-seconds timed out, whereas the optimal solution uses a direct mathematical math.comb formula.
>
> ```python
> # LeetCode 3422: Find the N-th Value After K Seconds
>
> ## LLM Solution
>
> class Solution:
>  def valueAfterKSeconds(self, n: int, k: int) -> int:
>   MOD = 10**9 + 7
>
>   # Initialize the array with all 1s
>   a = [1] * n
>
>   # Simulate k seconds
>   for _ in range(k):
>    # Create a new array for the next state
>    new_a = [0] * n
>    new_a[0] = a[0] # a[0] always remains 1
>
>    # Update each element based on the rule
>    for i in range(1, n):
>     # Sum of all preceding elements plus itself
>     new_a[i] = (sum(a[:i+1])) % MOD
>
>    # Update the array for the next iteration
>    a = new_a
>
>   # Return the value of a[n-1] after k seconds
>   return a[n-1]
>
> ## Canonical Solution
>
> class Solution:
>  def valueAfterKSeconds(self, n: int, k: int) -> int:
>   import math
>   return math.comb(k + n - 1, k) % 1_000_000_007
> ```
>
>
> In addition, we analyzed performance across different problem types. The model excels at procedural tasks like string-matching (MP=90.91) and simulation but struggles with problems requiring deeper reasoning, such as dynamic programming (MP=23.09), greedy algorithms, and combinatorics.
>
> *TABLE 4: Algorithm Tag Analysis*
>
> | Metric | string-matching | simulation | counting | hash-table | two-pointers | enumeration | string | sliding-window | math | sorting | array | graph | number-theory | binary-search | prefix-sum | matrix | bit-manipulation | heap-priority-queue | combinatorics | greedy | dynamic-programming |
> | :--- | :---: | :---: | :---: | :---: | :---: | :---: | :---: | :---: | :---: | :---: | :---: | :---: | :---: | :---: | :---: | :---: | :---: | :---: | :---: | :---: | :---: |
> | **$MP$ (%)** | 90.91 | 89.29 | 82.74 | 76.75 | 76.14 | 72.19 | 71.56 | 69.23 | 68.53 | 68.52 | 67.31 | 60.44 | 60.00 | 58.21 | 54.41 | 53.30 | 53.13 | 46.67 | 44.64 | 43.08 | 23.09 |
> | **$ET$ (%)** | 86.85 | 80.37 | 76.85 | 71.61 | 79.85 | 71.52 | 67.85 | 59.91 | 65.65 | 64.86 | 63.91 | 58.96 | 58.87 | 52.26 | 50.17 | 52.59 | 51.98 | 44.89 | 39.43 | 40.12 | 21.69 |
> | **$MI$ (%)** | 86.48 | 77.99 | 74.72 | 70.16 | 77.02 | 70.16 | 67.08 | 58.52 | 63.72 | 63.54 | 62.19 | 58.20 | 57.64 | 51.43 | 48.88 | 51.08 | 50.57 | 43.49 | 36.17 | 40.00 | 20.75 |
>
> We will add this new analysis to the paper. Thank you for helping us strengthen this aspect of our work.
>
>
>
> **D1: The data structure of the evaluation dataset is not clarified. Especially since the instructions are not specified for different LLMs, the generation results of each LLM may not be reproducible.**
>
> We thank the reviewer for this insightful comment. To address this, we have made two key additions to our manuscript, which directly answer both parts of the reviewer's concern. First, as detailed in our response to **W1**, we have now included a comprehensive **Dataset Card** (Table 1) and a **task distribution summary** (Table 2). The Dataset Card explicitly defines every field in our benchmark (`id`, `description`, `source`, `solutions`, `test_case_generator`, etc.), providing a clear and complete picture of the data's structure.
>
> Next, as explained in our response to **W2**, we use a single, unified instruction template for all models to ensure a fair and unbiased comparison. To further guarantee reproducibility, we provide a minimal working example below. This Python snippet demonstrates the exact procedure for loading a problem, formatting the prompt, and generating a solution with the specified parameters used in our study.
>
> ```python
> import torch
> from datasets import load_dataset
> from transformers import AutoTokenizer, AutoModelForCausalLM
>
> # 1. Load the model and tokenizer
> # This example uses deepseek-coder-1.3b-instruct, but the process is analogous for other models.
> model_id = "deepseek-ai/deepseek-coder-1.3b-instruct"
> tokenizer = AutoTokenizer.from_pretrained(model_id, trust_remote_code=True)
> model = AutoModelForCausalLM.from_pretrained(model_id, trust_remote_code=True, torch_dtype=torch.bfloat16).cuda()
>
> # 2. Load the EffiBench-X dataset from Hugging Face
> # Replace with the final Hugging Face dataset name upon publication.
> dataset = load_dataset(EffiBench-X Hugging Face Link)
> example = dataset["train"][0] # Get a sample problem
>
> # 3. Prepare the standardized prompt
> # This template is used for all models to ensure fair evaluation.
> task_description = example['description']
> target_lang = "Python" # Specify the target programming language
>
> prompt = f"""You are a competitive programming expert skilled in multiple programming languages.
> Your task is to solve the following problem in {target_lang}:
>
> {description}
> """
> inputs = tokenizer(prompt, return_tensors="pt").to(model.device)
>
> # 4. Generate the solution
> # We use temperature=0 for deterministic and reproducible outputs.
> outputs = model.generate(
>     **inputs,
>     max_new_tokens=2048,
>     temperature=0.0,
>     do_sample=False # Explicitly disable sampling
> )
> solution = tokenizer.decode(outputs[0], skip_special_tokens=True)
>
> print(solution)
> ```

---

> > ### Comment · Reviewer_rddN · 2025-08-09
> > **Official Comment**
> >
> > Thanks for the detailed responses. My concerns are almost addressed. I will raise the score accordingly.

---

### Note · Authors · 2025-08-14

We sincerely thank the AC and Reviewers for their thoughtful feedback and constructive input. Below, we summarize the major improvements made and key reviewer concerns addressed:

**Major Rebuttal Discussions & Resolutions**

1. **Dataset Transparency & Reproducibility** (Reviewer rddN):
 We added a **detailed dataset card and task distribution table**, along with a full prompt + code usage example to ensure reproducibility.

2. **Instruction Template Clarification** (Reviewer rddN):
 Clarified our use of a single standardized prompt for all models to ensure fair comparison and **eliminate prompt-induced bias**.

3. **Systematic Inefficiency Analysis** (Reviewers rddN, RKmc, 2u2z):
 We conducted a new large-scale analysis categorizing inefficiencies into **algorithmic, implementation-level, and I/O-related issues**, identifying semantic errors as dominant across all languages.

4. **Data Contamination Analysis** (Reviewer RKmc):
 We performed **n-gram** and **embedding-based** overlap analysis, confirming **minimal contamination (<1%)**.

5. **Advanced Prompting and Feedback Loops** (Reviewer 2u2z):
 We evaluated both **single-shot prompting** strategies and **iterative feedback refinement** (EffiLearner), showing significant gains in efficiency.

6. **Hardware Robustness** (Reviewer RKmc):
 We validated model rankings and metric stability across AMD and Intel CPUs, confirming hardware-agnostic design.

7. **Multi-Sample Evaluation (pass@5)** (Reviewer MDVc):
 We explored non-greedy decoding with temperature-based sampling. Selecting the **best of 5** correct generations improves efficiency while preserving model ranking.

8. **Prior Work (CodeScope) Comparison** (Reviewer MDVc):
 We acknowledged and discussed CodeScope, highlighting key differences in **scale, language diversity, and data contamination control**.

**Reviewer-Specific Responses & Improvements**

1. **Reviewer rddN**: Added dataset structure details, unified instruction templates, and inefficiency categorization by cause and algorithm type.

2. **Reviewer RKmc**: Provided contamination analysis, test generator metrics, unclipped score breakdown, cross-hardware comparison, and roadmap for system-level expansion.

3. **Reviewer MDVc**: Added multi-sample decoding results, clarified expert solution selection, and discussed novelty vs. CodeScope.

4. **Reviewer 2u2z**: Conducted deeper scientific analysis of LLM inefficiencies and introduced adaptive prompting and iterative refinement methods.

---

### Decision · Program_Chairs · 2025-09-18

**Decision:**

Accept (poster)

**Comment:**

Summary:

This paper introduces EffiBench-X, a multilingual benchmark designed to evaluate the efficiency of code generated by large language models (LLMs) across six programming languages. The benchmark consists of 623 competitive programming problems, each paired with a human-expert canonical solution, a test-case generator, and a language-specific harness, all executed in a docker-based sandbox with high-resolution profiling. This paper measures efficiency using three metrics: execution time (ET), memory peak (MP), and memory integral (MI). By evaluating 26 open- and closed-source models, this work finds that even the best model (Qwen3-32B) achieves only about 62% of human execution-time efficiency, highlighting a substantial gap between human and LLM-generated code.

Strengths:

- The reviewers have highlighted that this work presents a large-scale, multilingual benchmark focused on the efficiency (not just correctness) of LLM-generated code, spanning six languages and harder problems than prior datasets.
- The proposed benchmark uses a carefully engineered sandbox with low variance, automatic and manual validation of test cases, and transparent documentation of environments and settings.
- The problem is drawn from competitive programming with human-expert baselines, measures curb data contamination, and the paper clearly illustrates inefficiency patterns across languages and models.

Weaknesses:

- One of the key weaknesses was the lack of clarity and depth in the experimental design and presentation, including insufficient description of evaluation cases and instructions, as well as an overly simplistic reliance on tables that limit the interpretability of the results. This has been addressed in the rebuttal.
- Another weakness raised by the reviewer includes a lack of sufficient depth and breadth to fully establish the robustness, fairness, and practical relevance of the benchmark, as key aspects of data integrity, measurement design, execution variability, and real-world applicability were either underexplored or omitted. The authors provided a nice rebuttal, and the reviewer was satisfied with the response, provided all the changes are included in the final version.
- One reviewer also mentioned that paper contributions are weakened by missing prior work, limited evaluation methods, and unclear baseline choices. This concern was also addressed during the rebuttal, and the reviewer raised the scores.

Overall, this is a good contribution and all the reviewers recommended acceptance.